# Analysis of the genomic architecture of a complex trait locus in hypertensive rat models links *Tmem63c* to kidney damage

Angela Schulz[1,2†], Nicola Victoria Müller[1,2,3†], Nina Anne van de Lest[4], Andreas Eisenreich[1,2], Martina Schmidbauer[1,2], Andrei Barysenka[5], Bettina Purfürst[6], Anje Sporbert[7], Theodor Lorenzen[2], Alexander M Meyer[2], Laura Herlan[2], Anika Witten[5], Frank Rühle[5], Weibin Zhou[8], Emile de Heer[4], Marion Scharpfenecker[4], Daniela Panáková[9*], Monika Stoll[5,10‡], Reinhold Kreutz[2,9‡*]

[1]Charité - Universitätsmedizin Berlin, corporate member of Freie Universität Berlin, Humboldt-Universität zu Berlin, Berlin, Germany; [2]Institute of Clinical Pharmacology and Toxicology, Berlin Institute of Health, Berlin, Germany; [3]Max Delbrück Center for Molecular Medicine in the Helmholtz Association, Electrochemical Signaling in Development and Disease, Berlin, Germany; [4]Department of Pathology, Leiden University Medical Center (LUMC), Leiden, The Netherlands; [5]Westfälische Wilhelms University, Genetic Epidemiology, Institute for Human Genetics, Münster, Germany; [6]Max Delbrück Center for Molecular Medicine in the Helmholtz Association, Core Facility Electron Microscopy, Berlin, Germany; [7]Max Delbrück Center for Molecular Medicine in the Helmholtz Association, Advanced Light Microscopy, Berlin, Germany; [8]Division of Nephrology, Department of Medicine, Center for Human Disease Modeling, Duke University School of Medicine, Durham, United States; [9]DZHK (German Centre for Cardiovascular Research), Partner site Berlin, Berlin, Germany; [10]Department of Biochemistry, Maastricht University, Genetic Epidemiology and Statistical Genetics, Maastricht, The Netherlands

*For correspondence:
daniela.panakova@mdc-berlin.de (DP);
reinhold.kreutz@charite.de (RK)

†These authors contributed equally to this work
‡These authors also contributed equally to this work

Competing interests: The authors declare that no competing interests exist.

**Abstract** Unraveling the genetic susceptibility of complex diseases such as chronic kidney disease remains challenging. Here, we used inbred rat models of kidney damage associated with elevated blood pressure for the comprehensive analysis of a major albuminuria susceptibility locus detected in these models. We characterized its genomic architecture by congenic substitution mapping, targeted next-generation sequencing, and compartment-specific RNA sequencing analysis in isolated glomeruli. This led to prioritization of transmembrane protein *Tmem63c* as a novel potential target. *Tmem63c* is differentially expressed in glomeruli of allele-specific rat models during onset of albuminuria. Patients with focal segmental glomerulosclerosis exhibited specific TMEM63C loss in podocytes. Functional analysis in zebrafish revealed a role for *tmem63c* in mediating the glomerular filtration barrier function. Our data demonstrate that integrative analysis of the genomic architecture of a complex trait locus is a powerful tool for identification of new targets such as *Tmem63c* for further translational investigation.
DOI: https://doi.org/10.7554/eLife.42068.001

## Introduction

The analysis of the genetic basis of common diseases remains challenging due to their complex pathogenesis and genetic heterogeneity in human populations (*Deng, 2015*; *Glazier et al., 2002*;

**eLife digest** The human kidneys filter the entire volume of the blood about 300 times each day. This ability depends on specialized cells, known as podocytes, which wrap around some of the blood vessels in the kidney. These cells control which molecules leave the blood based on their size. Normally large molecules like proteins are blocked, while smaller molecules including waste products, toxins, excess water and salts pass through into the urine.

If this filtration system is damaged, by high blood pressure, for example, it can lead to chronic kidney disease. A hallmark of this disease, often called CKD for short, is high levels of the protein albumin in the urine. Previous studies involving rats with high blood pressure have found several regions of the genome that contribute to high levels of albumin in the urine, including one on chromosome 6. However, this region contains several genes and it was unclear which genes affected the condition.

Schulz et al. set out to narrow down the list and find specific genes that might contribute to elevated albumin in the urine of rats with high blood pressure. This search identified the gene for a protein called TMEM63c as a likely candidate. This protein spans the outer membrane of podocyte cells. Analysis of kidney biopsies showed that patients with chronic kidney disease also had low levels of this protein in their podocytes. Further experiments, this time in zebrafish, showed that reducing the activity of the gene for *tmem63c* led to damaged podocytes and a leakier filter in the kidneys.

The results suggest that this gene plays an important role in the integrity of the kidneys filtration barrier. It is possible that faulty versions of this gene are behind some cases of chronic kidney disease. If this proves to be the case, a better understanding of the role of this gene may lead to new treatments for the condition.

DOI: https://doi.org/10.7554/eLife.42068.002

*McCarthy et al., 2008*). This applies also to elevated blood pressure (BP) or hypertension, as recent meta-analyses of genome-wide association studies (GWAS) identified more than 100 gene loci associated with BP (*Hoffmann et al., 2017*; *Warren et al., 2017*). The effect size of the identified gene loci, however, is in general rather modest and less than 4% of the variance of BP phenotypes can be explained by these loci (*Hoffmann et al., 2017*; *Warren et al., 2017*), while around 40% to 50% of the variability appears heritable (*Levy et al., 2007*; *Miall and Oldham, 1963*). Considerable evidence supports a major role of the kidney in BP regulation, and for the renal damage such as albuminuria development as a consequence of long-term BP elevation (*Coffman and Crowley, 2008*; *Mancia et al., 2013*). Interestingly, genetic risk scores deploying BP genetic variants predict also hypertensive target organ damage in the heart, cerebral vessels, and eye, while only little evidence exists for an effect on kidney damage (*Ehret et al., 2016*). In this regard, several inbred hypertensive rat models provide valuable complementary tools to uncover the genetics of kidney damage in hypertension (*Schulz and Kreutz, 2012*; *Yeo et al., 2015*). These hypertensive models belong to the large panel of inbred strains that have been generated for a range of physiological and disease phenotypes by selective breeding (*Atanur et al., 2013*). Of interest, compensatory alleles that protect against hypertensive organ damage have also been inadvertently co-selected in this process and explain the impressive resistance of some hypertensive strains against target organ damage (*Jacob, 2010*; *Jeffs et al., 1997*; *Rubattu et al., 1996*; *Schulz and Kreutz, 2012*; *Yeo et al., 2015*). Quantitative trait loci (QTL) mapping strategies took advantage of these findings by intercrossing hypertensive strains with contrasting kidney damage phenotypes (*Schulz and Kreutz, 2012*). It remains challenging, however, to identify the molecular changes at a complex trait QTL that account for gene-gene or gene-environment interactions, incomplete penetrance, and epigenetic inheritance (*Buchner and Nadeau, 2015*; *Deng, 2015*).

Here, we successfully overcome the obstacles of QTL mapping in rodents by analyzing two inbred hypertensive rat strains with contrasting kidney damage with albuminuria phenotypes. We used the Munich Wistar Frömter (MWF) rat as a suitable hypertensive model system to target the genetic basis of the albuminuria phenotype in contrast to the spontaneously hypertensive rat (SHR) representing a model that is protected against albuminuria development (*Schulz and Kreutz, 2012*;

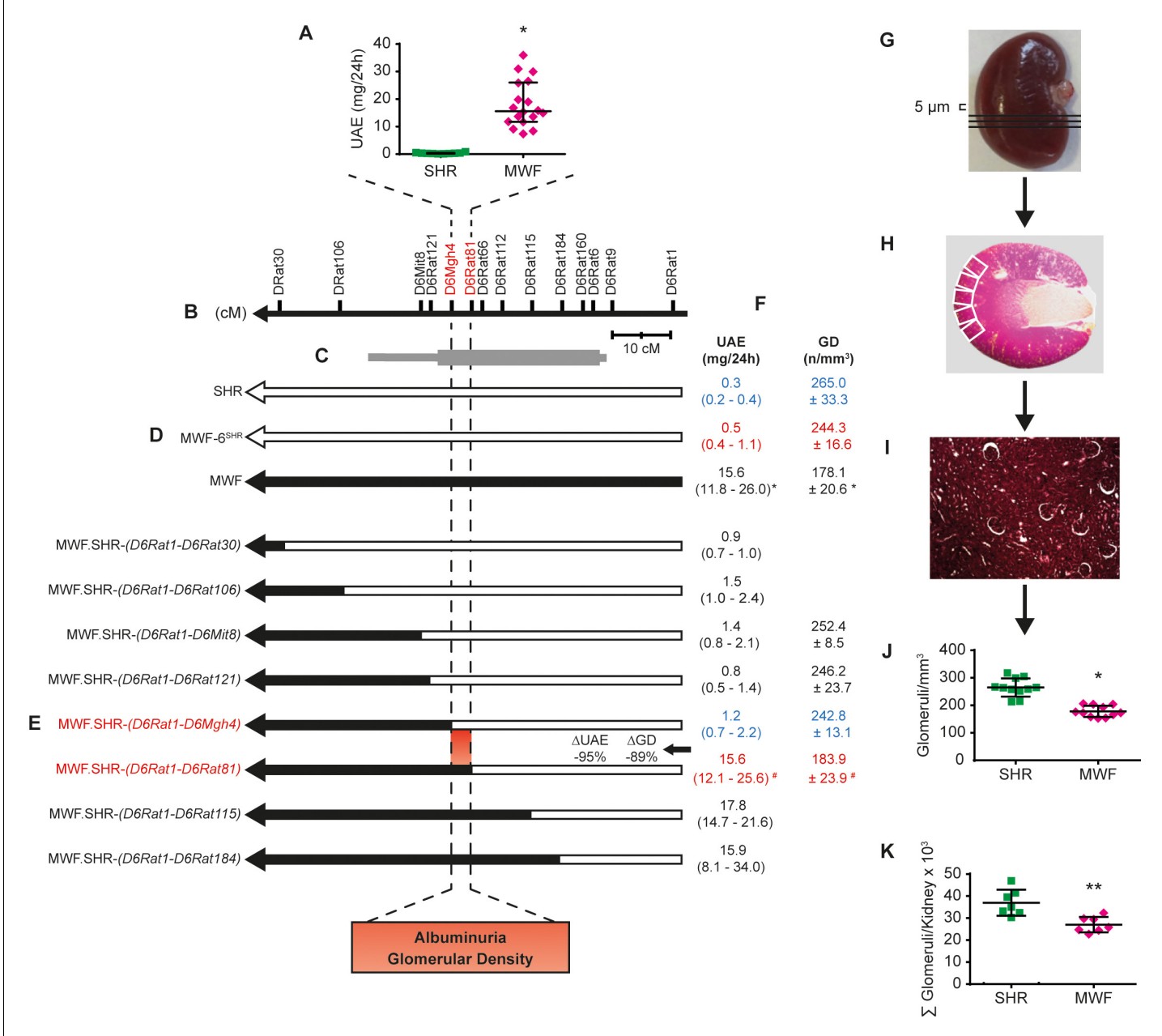

**Figure 1.** Congenic substitution mapping of the albuminuria and nephron deficit QTL on rat chromosome 6. (**A**) Urinary albumin excretion (UAE) in Munich Wistar Frömter (MWF) and spontaneously hypertensive rats (SHR) at 8 weeks of age. MWF ($n = 18$); SHR ($n = 10$); values plotted: median ±interquartile range (IQR); Mann-Whitney U test; *p<0.0001. (**B**) Genetic map of RNO6 with genetic markers and distance in centi Morgan (cM). (**C**) 1-LOD (thick bar) and 2-LOD (thin bar) confidence intervals for placement of the albuminuria QTL by linkage mapping. (**D**) The chromosomal fragment for the MWF (black) and SHR (white) genome are indicated for the MWF ($n = 18/11^§$), consomic MWF-6$^{SHR}$ ($n = 19/11^§$), and SHR ($n = 10/11^§$) strains. (**E–F**) The chromosomal fragment for congenic strains designated as MWF.SHR-(*D6Rat1-D6Rat184*) ($n = 17/0^§$), MWF.SHR-(*D6Rat1-D6Rat115*) ($n = 10/0^§$), MWF.SHR-(*D6Rat1-D6Rat81*) ($n = 25/11^§$), MWF.SHR-(*D6Rat1-D6Mgh4*) ($n = 24/11^§$), MWF.SHR-(*D6Rat1-D6Rat121*) ($n = 29/6^§$), MWF.SHR-(*D6Rat1-D6Mit8*) ($n = 18/6^§$), MWF.SHR-(*D6Rat1-D6Rat106*) ($n = 10/0^§$), and MWF.SHR-(*D6Rat1-D6Rat30*) ($n = 10/0^§$) (**E**). Corresponding phenotypes for UAE and glomerular density (GD) (**F**). Red values indicate disease phenotypes and blue values an amelioration of phenotypes for informative strains. $^§$ n is presented for the phenotypes in the following order UAE/GD; values shown for UAE: median ±IQR, Kruskall-Wallis test with Dunn's multiple comparisons test; *p<0.0001; values shown for GD: mean ±SD; one-way ANOVA with post hoc Bonferroni's multiple comparisons test and Mann-Whitney U test; *p<0.0001 vs. SHR, MWF-6$^{SHR}$, MWF.SHR-(*D6Rat1-D6Mgh4*), respectively; $^#$p<0.0001 vs. MWF.SHR-(*D6Rat1-D6Mgh4*), respectively; $^{##}$p=0.029 vs. MWF.SHR-(*D6Rat1-D6Rat30*). (**G–K**) Evaluation of nephron deficit in MWF and SHR rats by determination of GD. Right kidneys at 4 weeks of age were cut into 5 μm thick histological sections (**G**). For evaluation of glomerular diameter and glomerular number adjacent pictures were taken for each periodic acid-Schiff (PAS) stained histological section (white rectangles) (**H**). Calculation of GD (10x magnification) (**I**). Direct comparison of GD

*Figure 1 continued on next page*

*Figure 1 continued*

evaluation (J) vs. total glomerular number as previously estimated by the physical fractionator method (K). Total glomerular number ($n = 7$ each); GD ($n = 11$ each); values plotted: mean ±SD; two-tailed student's t-test; *p<0.0001 vs. SHR; **p=0.0024 vs. SHR.

DOI: https://doi.org/10.7554/eLife.42068.003

The following source data is available for figure 1:

**Source data 1.** Albuminuria in parental, consomic, and congenic rat strains.

DOI: https://doi.org/10.7554/eLife.42068.004

**Source data 2.** Glomerular density and total nephron number in parental, consomic, and congenic rat strains.

DOI: https://doi.org/10.7554/eLife.42068.005

*van Es et al., 2011*). Following initial QTL mapping in intercrosses of contrasting hypertensive strains (*Schulz and Kreutz, 2012*), we successfully combined conventional fine mapping using congenic strains with characterization of the genomic architecture in sub-QTLs by targeted next-generation sequencing (NGS), and compartment-specific RNA sequencing (RNA-Seq) analysis. This comprehensive approach allowed us to prioritize transmembrane protein *Tmem63c* as a novel positional target for albuminuria development among several candidates. Functional relevance of *Tmem63c* was supported by allele-specific and BP-independent differential glomerular expression during onset of albuminuria in allele-specific minimal congenic rat lines derived from the contrasting rat models. Loss of glomerular TMEM63C expression in podocytes of patients with focal segmental glomerulosclerosis (FSGS) provided strong evidence for translational relevance. Loss-of-function studies in zebrafish induced a glomerular filtration barrier (GFB) defect compatible with the albuminuria phenotype, which was rescued upon co-injection of zebrafish *tmem63c* mRNA or rat *Tmem63c* mRNA, showing not only the specificity of the observed knockdown phenotype, but also conservation of the gene's function across species. Ultrastructural analysis by electron microscopy, demonstrated severe morphological defects including podocyte damage with foot process effacement in *tmem63c-*

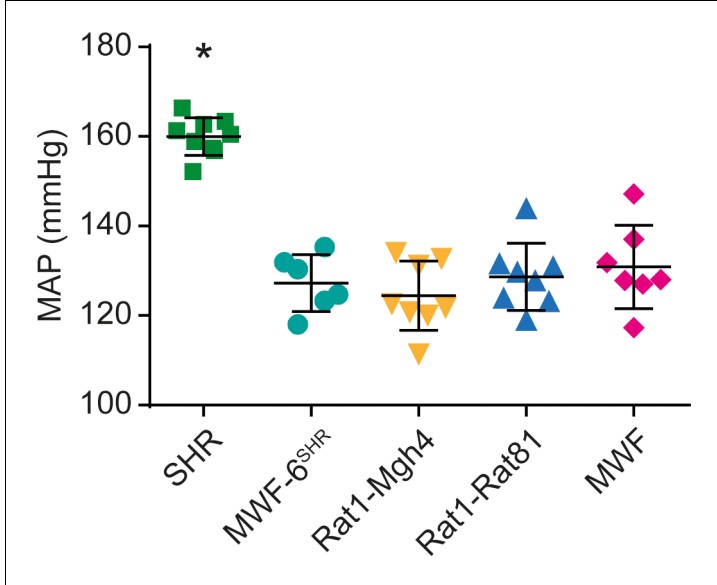

**Figure 2.** Mean arterial blood pressure (MAP) in consomic and congenic rat strains. Measurement of MAP is shown for SHR ($n = 9$), MWF-6[SHR] ($n = 6$), congenic MWF.SHR-(*D6Rat1-D6Mgh4*) (Rat1-Mgh4, $n = 8$) and congenic MWF.SHR-(*D6Rat1-D6Rat81*) (Rat1-Rat81, $n = 8$) and MWF ($n = 7$) at week 14; values plotted: mean ±SD; one-way ANOVA with Bonferroni's post hoc analysis; *p<0.0001 vs. other strains, respectively.

DOI: https://doi.org/10.7554/eLife.42068.006

The following source data is available for figure 2:

**Source data 1.** Mean Arterial Pressure in parental, consomic, and congenic rat strains.

DOI: https://doi.org/10.7554/eLife.42068.007

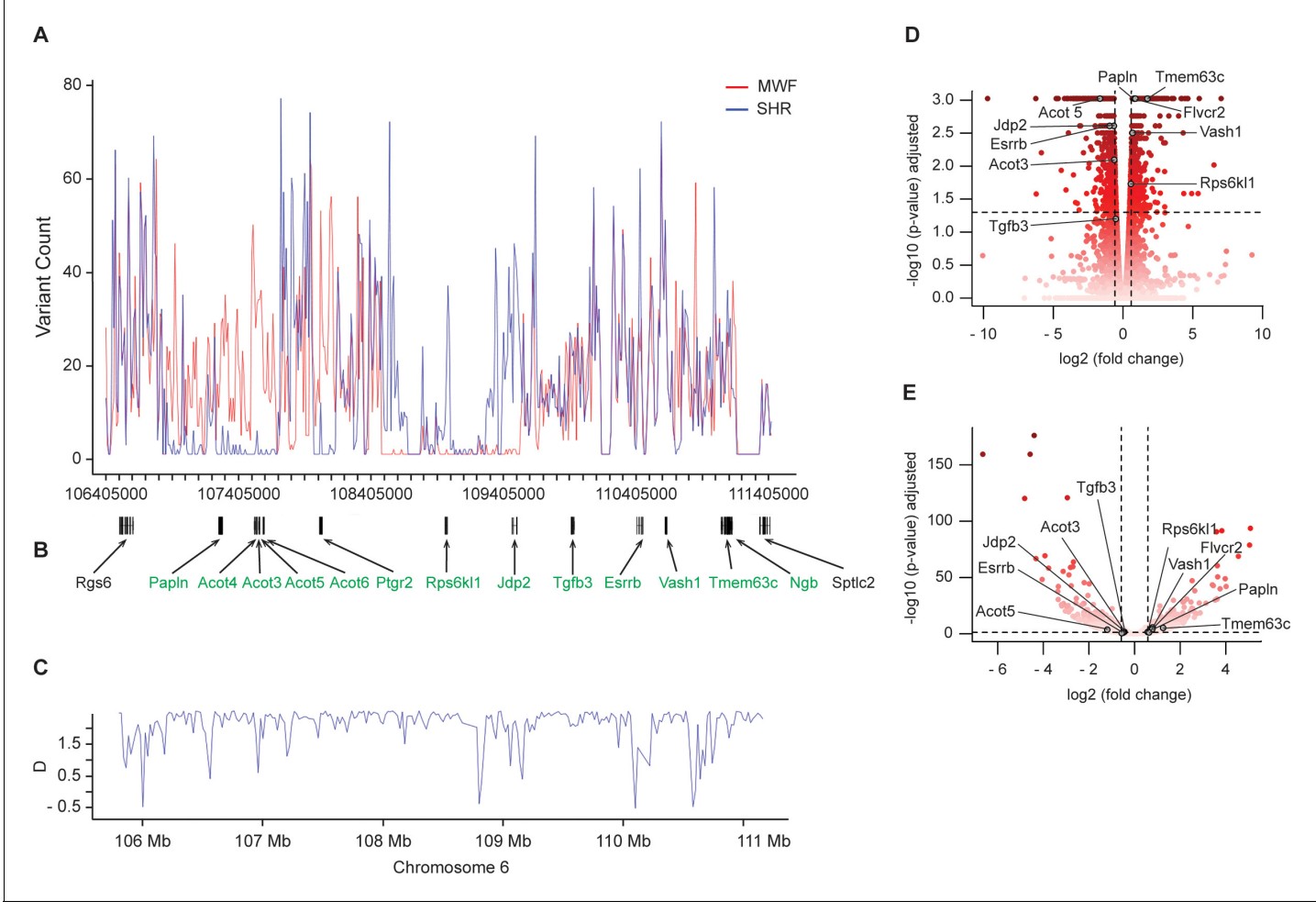

**Figure 3.** Next-generation sequencing (NGS) analysis of the kidney damage region on rat chromosome 6 and RNA sequencing (RNA-Seq) analysis in isolated glomeruli. (A–B) DNA sequencing (DNA-Seq) analysis revealed the numbers of DNA variants in comparison to the reference genome across the sequenced region for the Munich Wistar Frömter (MWF) and spontaneously hypertensive rat (SHR) strains ($n = 3$, each) (A). The physical map between nucleotide position 106,405,000 and 111,405,000 is shown together with positions of potential candidate genes (in green) (B); in addition, the two genes at the 5'-position and 3'-position of the candidate region are visualized in black. (C) The Tajima's D variation across the sequenced region. (D–E) Volcano plots illustrating the differential expression results (MWF vs. SHR) in RNA-Seq analysis in isolated glomeruli using the Cuffdiff (D) and DESeq2 (E) analysis tools. For each gene, the log10 transformed differential expression *P*-value adjusted for false discovery rate is plotted against the log2 transformed expression fold change. The color gradient refers to the P-values given on the y-axis. The applied significance threshold of adjusted p-value<0.05 is indicated as dashed horizontal line. Dashed vertical lines indicate fold changes > 1.5. Genes of interest are highlighted in grey and are annotated with gene symbols. (See *Figure 3—figure supplement 1*).

DOI: https://doi.org/10.7554/eLife.42068.008

The following source data and figure supplement are available for figure 3:

**Source data 1.** Predicted effects of variants identified by NGS in the candidate region on rat chromosome 6 in human.
DOI: https://doi.org/10.7554/eLife.42068.010

**Figure supplement 1.** Venn diagram of NGS analysis.
DOI: https://doi.org/10.7554/eLife.42068.009

deficient embryos. Altogether, our findings reveal TMEM63C being integral to podocyte physiology, and identified its potential role in glomerular renal damage in patients with chronic kidney disease. Our data demonstrate, despite the difficulties of QTL analysis in experimental models such as inbred strains, the feasibility to identify novel targets by combining conventional congenic substitution mapping with integrative analysis of genomic architecture of identified susceptibility loci and functional studies.

**Table 1.** Significant variants in the candidate kidney damage region on rat chromosome 6.

| Gene | Gene coordinates | | Variant position | Variant type | Allelic variants | | Amino acid exchange | Effect of sequence variant | PROVEAN score |
|------|------------------|---|------------------|--------------|------------------|---|---------------------|----------------------------|---------------|
| | Start position (bp) | Stop position (bp) | | | MWF | SHR | | | |
| Acot4 | 107,517,668 | 107,522,952 | 107,518,131 | exonic | A | G | Gly → Arg | non-synonymous | −5.660 |
| Acot5 | 107,550,904 | 107,557,688 | 107,551,446 | exonic | A | C | Arg → Ser | non-synonymous | −2.842 |
| | | | 107,551,528 | exonic | A | G | Arg → His | non-synonymous | −2.646 |
| | | | 107,551,717 | exonic | G | C | Pro → Arg | non-synonymous | −5.314 |
| | | | 107,557,092 | exonic | A | T | Leu → Gln | non-synonymous | −5.291 |
| Acot6 | 107,581,608 | 107,590,373 | 107,590,006 | exonic | C | T | Leu → Pro | non-synonymous | −5.091 |
| Ptgr2 | 108,009,251 | 108,029,859 | 108,029,833 | exonic | T | C | Arg → Cys | non-synonymous | −3.672 |
| Ngb | 111,126,261 | 111,132,320 | 111,128,730 | exonic | G | A | Leu → Pro | non-synonymous | −3.000 |
| | | | 111,131,291 | exonic | ACT | A | NA | frameshift deletion | NA |

MWF, Munich Wistar Frömter; SHR, spontaneously hypertensive rat. NA, not applicable.

DOI: https://doi.org/10.7554/eLife.42068.011

# Results

## Congenic substitution mapping in the MWF rat model refines the albuminuria QTL to a chromosomal region linked to both albuminuria and nephron deficit

We previously confirmed the pivotal role of a major albuminuria QTL on rat chromosome 6 (RNO6) in the MWF model by generating a consomic MWF-6[SHR] strain, which carries RNO6 from the contrasting albuminuria-resistant SHR strain in the MWF genetic background (*Figure 1A–D*) (*Schulz et al., 2007*). In addition, MWF rats inherit a deficit in nephron (and glomeruli) number, which represents a predisposition for the development of both hypertension and kidney damage (*Wang and Garrett, 2017*). We performed congenic substitution mapping for both albuminuria and glomerular density phenotypes by generating eight congenic lines by introgression of nested chromosomal fragments from SHR onto the MWF genetic background, and compared the renal

**Table 2.** Presence of the frameshift deletion in inbred rat strains in neuroglobin (*Ngb*) at 111,131,291 bp.

| Strain | Presence of deletion | Strain | Presence of deletion |
|--------|----------------------|--------|----------------------|
| ACI | no | MHS | no |
| BBDP | no | MNS | no |
| BN.Lx | no | SBH | no |
| EVE | no | SBN | no |
| F344/Ncrl | no | **SHR** | **yes** |
| FHH | no | **SHR/NHsd** | **yes** |
| FHL | no | **SHRSP/Gla** | **yes** |
| **GK** | **yes** | SR/Jr | no |
| LE/Stm | no | SS/Jr | no |
| LEW | no | SS_JRHSDMCWI | no |
| LEW/NcrlBR | no | WAG | no |
| LH | no | **WKY** | **yes** |
| LL | no | **WKY/Gla** | **yes** |
| LN | no | **WKY_NHSD** | **yes** |

The inbred rat strains with presence of the *Ngb* frame shift deletion in bold belong to a clade of Wistar rat derived strains from Japan.

DOI: https://doi.org/10.7554/eLife.42068.012

**Table 3.** Genes in the candidate kidney damage region with differential expression in RNA-Seq analysis.

| Gene | Start (bp position) | End (bp position) | ID | P(DeSeq2) | P(Cuffdiff) | Strain[a] |
|------|---------------------|-------------------|-----|-----------|-------------|---------|
| Acot3 | 107,531,528 | 107,536,789 | ENSRNOG00000053460 | 0.09426742 | 0.00799931 | SHR |
| Rps6kl1 | 108,961,994 | 108,976,489 | ENSRNOG00000005530 | 0.024515465 | 0.0186294 | MWF |
| **Acot5** | 107,550,904 | 107,557,688 | ENSRNOG00000032508 | 0.000234652 | 0.000948356 | SHR |
| **Papln** | 107,245,820 | 107,276,755 | ENSRNOG00000009448 | 8.82E-05 | 0.000948356 | MWF |
| **Tmem63c** | 111,049,559 | 111,120,799 | ENSRNOG00000011334 | 6.26E-06 | 0.000948356 | MWF |
| **Flvcr2** | 109,617,355 | 109,681,495 | ENSRNOG00000008754 | 2.06E-06 | 0.000948356 | MWF |
| Jdp2 | 109,466,060 | 109,505,161 | ENSRNOG00000008224 | 0.052146312 | 0.00245615 | SHR |
| Esrrb | 110,410,141 | 110,455,906 | ENSRNOG00000010259 | 0.278291584 | 0.00245615 | SHR |
| Tgfb3 | 109,913,757 | 109,935,533 | ENSRNOG00000009867 | 0.019914732 | 0.0632265 | SHR |
| Vash1 | 110,624,856 | 110,637,382 | ENSRNOG00000010457 | 0.070552875 | 0.00314268 | MWF |

MWF, Munich Wistar Frömter; RNA-Seq, RNA sequencing; SHR, spontaneously hypertensive rat. Genes shown in bold were found to be significantly differentially expressed using both CuffDiff and DeSeq2 analysis. [a] The strain name is given for upregulation of mRNA expression.

DOI: https://doi.org/10.7554/eLife.42068.016

phenotypes between congenic lines and the parental MWF strain (*Figure 1E,F*). We successfully narrowed the original region identified by QTL mapping, spanning about 55 Mb between genetic markers *D6Rat106* and *D6Rat9* (*Schulz et al., 2003*), to a smaller interval comprising 4.9 Mb between *D6Mgh4* and *D6Rat81* (*Figure 1E,F*). The comparison between the two informative congenic lines MWF.SHR-(*D6Rat1-D6Mgh4*) and MWF.SHR-(*D6Rat1-D6Rat81*) revealed that 95% of the difference in albuminuria and 89% of the difference in glomerular density observed between the MWF and consomic MWF-6^SHR strains is attributable to this interval (*Figure 1E,F*). Subsequently, we set out to analyze the BP phenotype by direct intra-arterial measurements in the two congenic lines in comparison to the MWF and SHR strains. This analysis revealed similar mean arterial BP values in the two congenic lines and parental MWF strain (*Figure 2*). This findings clearly indicate the UAE difference between MWF.SHR-(*D6Rat1-D6Mgh4*) and MWF.SHR-(*D6Rat1-D6Rat81*) are not attributable to BP differences (*Figure 2*). Thus, we dissected away a role of this region for BP regulation and show that both the albuminuria and glomerular density phenotype co-localize in the same refined locus (sub-QTL), supporting further exploration of this region as an independent candidate region for kidney damage. Direct comparison of glomerular density (*Figure 1G–J*) with the total glomerular number as previously estimated by the physical fractionator method (*Figure 1K*) (*Schulz et al., 2007*) confirmed the nephron deficit in MWF compared to SHR (*Figure 1J*, p<0.0001). We further demonstrated that this phenotype maps also to RNO6 (*Figure 1F*) when determined by glomerular density analysis in kidney sections (*Figure 1G–J*).

## Targeted NGS analysis of the identified kidney damage candidate region

We employed targeted NGS on the region ranging from nucleotide positions 105,780,000 to 111,425,000 on RNO6. After genotype calling and comparing genotypes of consomic and congenic rats, we refined coordinates of the kidney damage locus from 106,400,000 bp to 111,360,000 bp (*Figure 3A,B*). This region contains 75 predicted protein-coding genes. In comparison to the reference genome (Rattus norvegicus, ENSEMBL rn6.0) (*Yates et al., 2016*), we identified 5,158 SNPs and 1893 small insertions and deletions (INDELs) in MWF, and 5326 single nucleotide polymorphisms (SNPs) and 1804 INDELs in SHR (*Figure 3—figure supplement 1*). Direct comparison between MWF and SHR revealed that both strains differ for 5376 SNPs and for 1613 INDELs (*Figure 3—figure supplement 1*), showing a remarkable pattern of stretches of variants either coming from the MWF or the contrasting SHR strain (*Figure 3A*). As selective sweeps have recently been reported as a consequence of inbreeding in rats (*Atanur et al., 2013*), we performed a formal analysis of selective processes such as directional selection or balancing selection, genetic hitchhiking, or introgression using the Tajima's D statistics (*Tajima, 1989*). We observe such traces of selection with the majority of Tajima's D values exceeding the generally accepted threshold of D > 2 (*Figure 3C*).

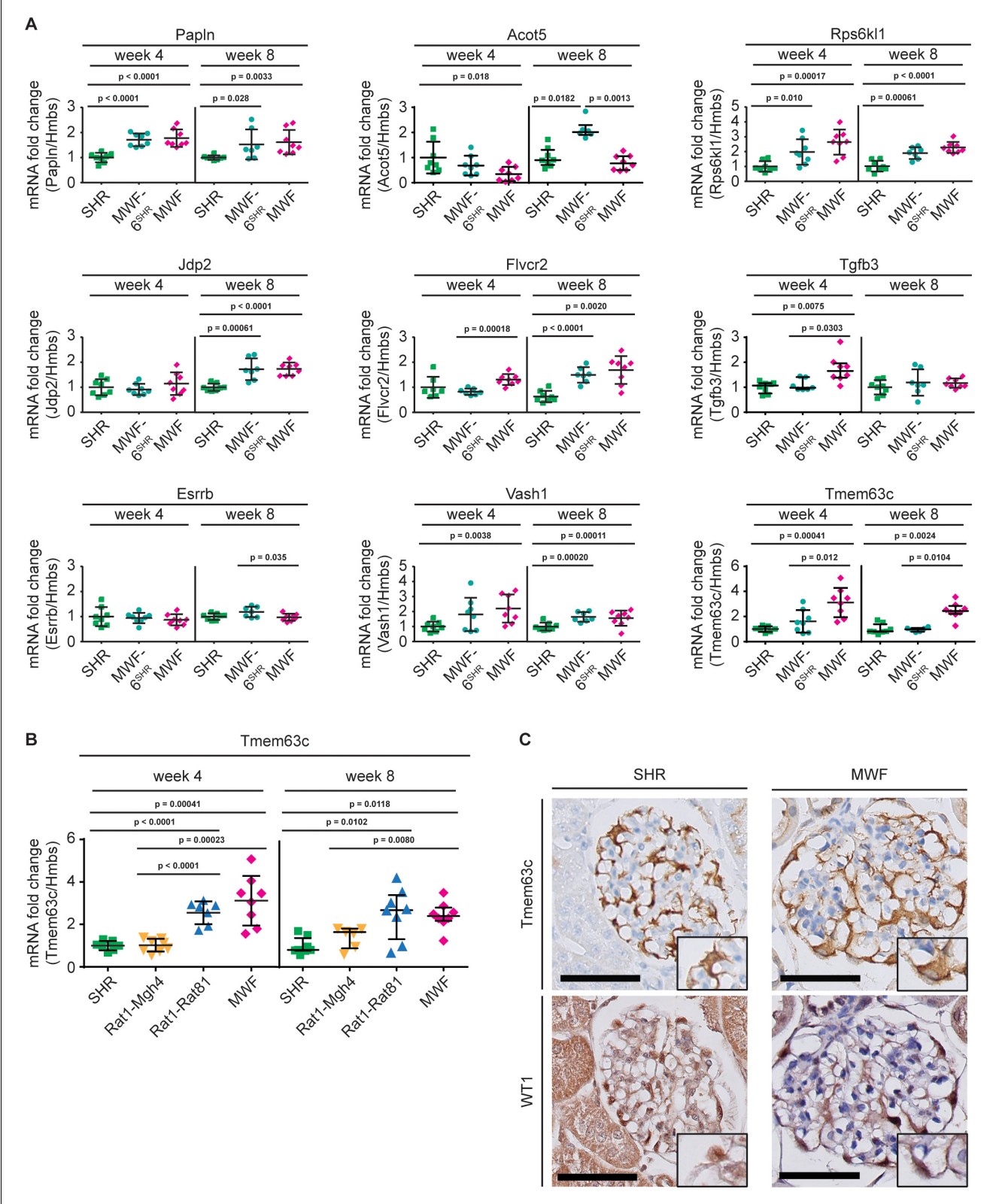

**Figure 4.** Validation of differentially expressed genes in isolated glomeruli by quantitative real-time PCR (qPCR) analysis. (**A**) qPCR analysis showed no consistent differential expression for eight genes from overall 10 genes identified with differential expression in RNA sequencing (RNA-Seq) analysis (*Table 3*) between parental Munich Wistar Frömter (MWF), spontaneously hypertensive rats (SHR), and consomic MWF-6[SHR] during the crucial time window for onset of albuminuria between weeks 4 and 8. Only transmembrane protein 63c (*Tmem63c*) demonstrated differential mRNA expression at

*Figure 4 continued on next page*

*Figure 4 continued*
both time points. Acyl-CoA thioesterase 3 (*Acot3*) (*Table 3*) showed very low mRNA expression in qPCR analysis precluding quantitative analysis. Consequently, *Acot3* and the genes shown were excluded from and *Tmem63c* was included for further functional analysis. Rats per strain (n = 7–8, each); values for *Acot5* (week 8),*Tgfb3* (week 4) and *Tmem63c* (week 8) are plotted as median ±IQR, while the rest of data are plotted as mean ±SD; data for *Acot5* (week 8), *Tgfb3* (week 4) and *Tmem63c* (week 8) were analyzed using Kruskal-Wallis test with Dunn's post-hoc analysis and Mann-Whitney U test, while the rest of data was analyzed by one-way ANOVA with Bonferroni's post hoc analysis and Mann-Whitney U test. (B) mRNA expression analysis for *Tmem63c* in isolated glomeruli by qPCR analysis is shown for MWF, SHR, congenic MWF.SHR-(*D6Rat1-D6Mgh4*) (Rat1-Mgh4) and congenic MWF.SHR-(*D6Rat1-D6Rat81*) (Rat1-Rat81), at week 4 and week 8. Rats per strain (n = 6–8, each); data for week 8 are plotted as median ±IQR and analyzed using Kruskal-Wallis test with Dunn's post hoc analysis and Mann-Whitney U test; the other data are plotted as mean ±SD and analyzed by one-way ANOVA with Bonferroni's post hoc analysis and Mann-Whitney U test (C) Representative immunohistochemical stainings of TMEM63C and Wilms tumor 1 (WT1) on kidney sections from SHR and MWF at 8 weeks of age; the insert indicates expression in podocytes. Scale bar = 50 μm. Quantitative analysis of TMEM63C intensity in podocytes using one-way ANOVA revealed lower intensity in MWF (n = 7) vs. SHR (n = 6) at 8 weeks (p=0.0032).
DOI: https://doi.org/10.7554/eLife.42068.013
The following source data is available for figure 4:

**Source data 1.** Quantitative real-time PCR analysis of differentially expressed genes in isolated glomeruli between rat strains.
DOI: https://doi.org/10.7554/eLife.42068.014
**Source data 2.** Primer list for quantitative real-time PCR analysis.
DOI: https://doi.org/10.7554/eLife.42068.015

Functional annotation using PROVEAN scores identified eight potentially deleterious non-synonymous variants in five genes (*Table 1*). In addition, in the contrasting SHR reference strain one frameshift deletion was detected in the gene encoding neuroglobin (*Ngb*). However, this deletion is also present in the entire clade of SHR-related rat strains derived from one ancestor including other strains with normal urinary albumin excretion and normal kidney function (*Table 2*) (*Atanur et al., 2013*). Consequently, this frameshift deletion is not to be considered involved in the kidney damage phenotype and was not further pursued. Thus, we identified no obvious single candidate by NGS analysis in the sub-QTL.

## Positional candidate gene identification by transcriptome analysis in isolated glomerular tissue by RNA-Seq

In order to identify other positional candidates at the sub-QTL on RNO6, we next embarked on RNA-Seq analysis in isolated glomeruli from MWF and SHR to assess global mRNA transcription patterns in the target compartment. We performed gene-based differential expression analysis using *Cuffdiff* and *DESeq2* software tools (*Figure 3D,E*). After correcting for multiple testing, 1838 genes were assigned a p-value<0.05 for *Cuffdiff* analysis and 1841 genes for *DESeq2*, respectively, yielding a total set of 2454 unique differentially expressed genes. When filtering the results for those genes residing in the candidate region, we identified a total of 10 genes to be significantly differentially expressed between MWF and SHR (*Table 3*, *Figure 3D,E*) at significant p-values. These genes were taken forward to validation by quantitative real-time PCR (qPCR) analysis in isolated glomeruli obtained from the two parental and MWF-6$^{SHR}$ consomic animals during onset of albuminuria occurring between 4 and 8 weeks of age (*Figure 4A*). This analysis revealed that from the nine genes, which could be analyzed, only *Tmem63c*, showed consistent and allele-dependent differential expression during the crucial time window (*Figure 4A,B*).

## *Tmem63c* represents a positional candidate gene in MWF with differential glomerular expression

RNA-Seq and qPCR analysis indicated a significant 2.5- to 3-fold upregulation of *Tmem63c* mRNA expression in isolated glomeruli in the MWF model, which was abolished by transfer of RNO6 from SHR onto the MWF genetic background in the corresponding MWF-6$^{SHR}$ consomic line (*Figure 4A*). Moreover, comparison of *Tmem63c* mRNA expression between the two informative congenic lines MWF.SHR-(*D6Rat1-D6Rat81*) and MWF.SHR-(*D6Rat1-D6Mgh4*) confirmed an allelic (cis) regulation of *Tmem63c* mRNA expression in MWF and SHR, and its association with albuminuria (*Figure 4B*). Further evaluation of the NGS data for *Tmem63c* in MWF and SHR revealed no significant sequence variants, with the exception of one detected variant in intron 18 (ENSRNOT00000015571) at

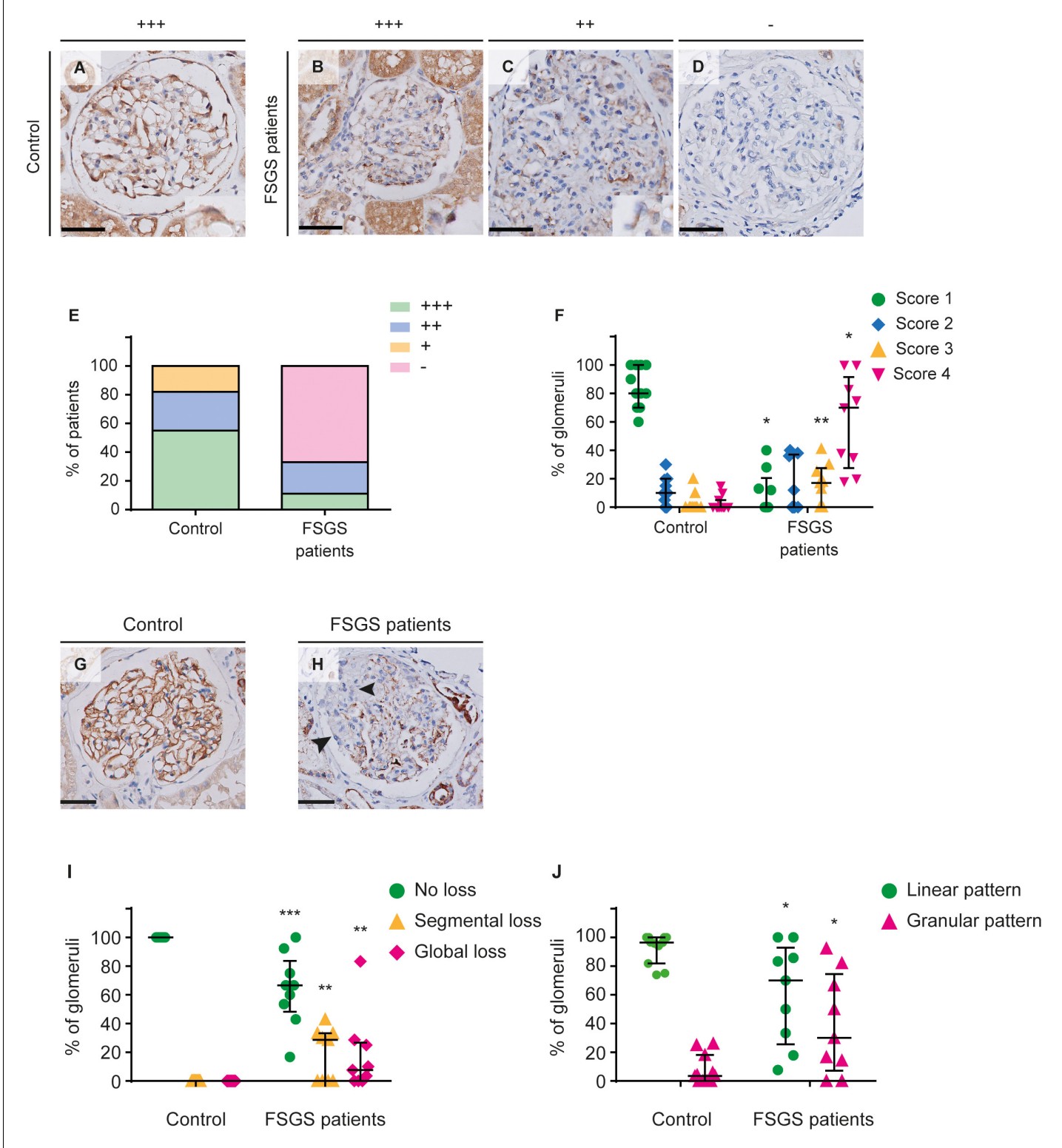

**Figure 5.** Transmembrane protein 63c (TMEM63C) protein and Nephrin expression deficiency in patients with focal segmental glomerulosclerosis (FSGS). (A–D) High-intensity TMEM63C staining in a glomerulus of a healthy human control subject (A) and representative glomeruli of FSGS patients with high (+++) (B), intermediate (+) TMEM63C staining intensity (C), or loss (-) of TMEM63C protein expression (D). The inserts indicate TMEM63C-positive podocytes. Scale bar = 50 µm. (E) Scoring of the TMEM63C staining intensity in controls (n = 11) and FSGS patients (n = 9). Percentage of cases with high intensity (green); intermediate intensity (blue); low intensity (yellow) and no TMEM63C staining (magenta). Linear-by-Linear association; p=0.005. (F) Scoring of the percentage of TMEM63C positivity in glomeruli. Green: no loss; blue:<25% loss; yellow: 25–50% loss; magenta:>50% loss of

*Figure 5 continued on next page*

Figure 5 continued

TMEM63C expression. Controls (n = 11) and FSGS patients (n = 9); values plotted: median ±IQR; Mann-Whitney U test; *p<0.0001 vs. control; **p=0.021 vs. control. (G–J) Linear nephrin staining in a glomerulus of a healthy human control subject (G) and segmental loss of nephrin staining in a glomerulus of a patient with FSGS (H), indicated by arrowheads. Nephrin expression was significantly reduced in patients with FSGS compared to healthy controls (I). Moreover, we observed a shift from a normal linear staining pattern, following the glomerular capillary wall, to a granular staining pattern (H and J). values plotted: median ±IQR; Mann-Whitney U test; *p=0.05 vs. control; **p<0.01 vs. control; ***p<0.001 vs. controls.

DOI: https://doi.org/10.7554/eLife.42068.017

The following source data is available for figure 5:

Source data 1. TMEM63C intensity score and percentage of TMEM63C positivity in glomeruli in FSGS patients.

DOI: https://doi.org/10.7554/eLife.42068.018

Source data 2. Nephrin staining in FSGS patients.

DOI: https://doi.org/10.7554/eLife.42068.019

111,101,251 bp at a potential splice site position. However, when analyzing our RNA-Seq data concerning differential exon usage in *Tmem63c*, we found no significant difference in exon usage between both parental strains.

We then set out to perform immunohistochemistry analysis of TMEM63C in MWF kidney. This revealed TMEM63C expression in a podocyte-specific pattern (*Figure 4C*). In contrast to the clearly elevated mRNA expression in isolated glomeruli, this analysis indicated no elevated glomerular protein expression in MWF, and only somewhat lower TMEM63C protein expression in podocytes in MWF compared to SHR at onset of albuminuria at 8 weeks of age (*Figure 4C*).

## Patients with focal segmental glomerulosclerosis exhibit loss of glomerular TMEM63C expression

Aging MWF rats develop histopathological changes similar to those observed in patients with FSGS (*Remuzzi et al., 1992*; *Schulz and Kreutz, 2012*). Podocyte injury with the development of glomerular proteinuria represents a pivotal hallmark of FSGS (*D'Agati et al., 2011*; *Lim et al., 2016*). Thus, we explored TMEM63C expression in patients with FSGS and healthy controls to evaluate its potential role for human kidney damage (*Yu et al., 2016*). This analysis demonstrated that TMEM63C is expressed in podocytes of all glomeruli in healthy controls (*Figure 5A,E*), while patients with FSGS exhibit a significant decrease of TMEM63C expression (*Figure 5B–E*) with a global loss of glomerular TMEM63C in the majority of patients analyzed (*Figure 5D–F*). In addition to TMEM63C expression, we analyzed the expression of nephrin protein as a pivotal component of the slit diaphragm of the GFB (*Figure 5G–J*) (*Kestilä et al., 1998*). Nephrin expression was also significantly reduced in patients with FSGS (*Figure 5H,I*), which is in accordance with previously published results (*Kim et al., 2002*). Moreover, we observed a shift from the normal linear staining pattern to a granular staining pattern as reported (*Figure 5H,J*) (*Doublier et al., 2001*; *Wernerson et al., 2003*).

## Knockdown of TMEM63C expression in human podocytes impairs cell viability and survival signaling

We further analyzed the effect of reduced *TMEM63C* expression in human podocytes in culture using small interfering RNA (siRNA) methodology (*Figure 6A*). We found significantly impaired cell viability in response to TMEM63C downregulation (*Figure 6B*). In addition, reduction of TMEM63C expression by siRNA decreased pro-survival signaling in human podocytes as indicated by reduced levels of pAKT (*Figure 6C*), and increased pro-apoptotic transition of cytochrome C from mitochondria to the cytoplasm (*Figure 6D*).

## Functional analysis of *tmem63c* in zebrafish

To assess the functional role of *tmem63c* for albuminuria development, we utilized the transgenic zebrafish line *Tg[fabp10a:gc-EGFP]* (*Zhou and Hildebrandt, 2012*). This model expresses a vitamin D binding protein tagged with enhanced green fluorescent protein (gc-EGFP) in the liver, from which it is released into the blood stream and circulates under the normal conditions in the blood (*Figure 7A,G*). Upon GFB damage, gc-EGFP leaks through the glomerular filtration barrier, indicated by a marked decrease in fluorescence in the trunk vasculature of *Tg[fabp10a:gc-EGFP]* embryos

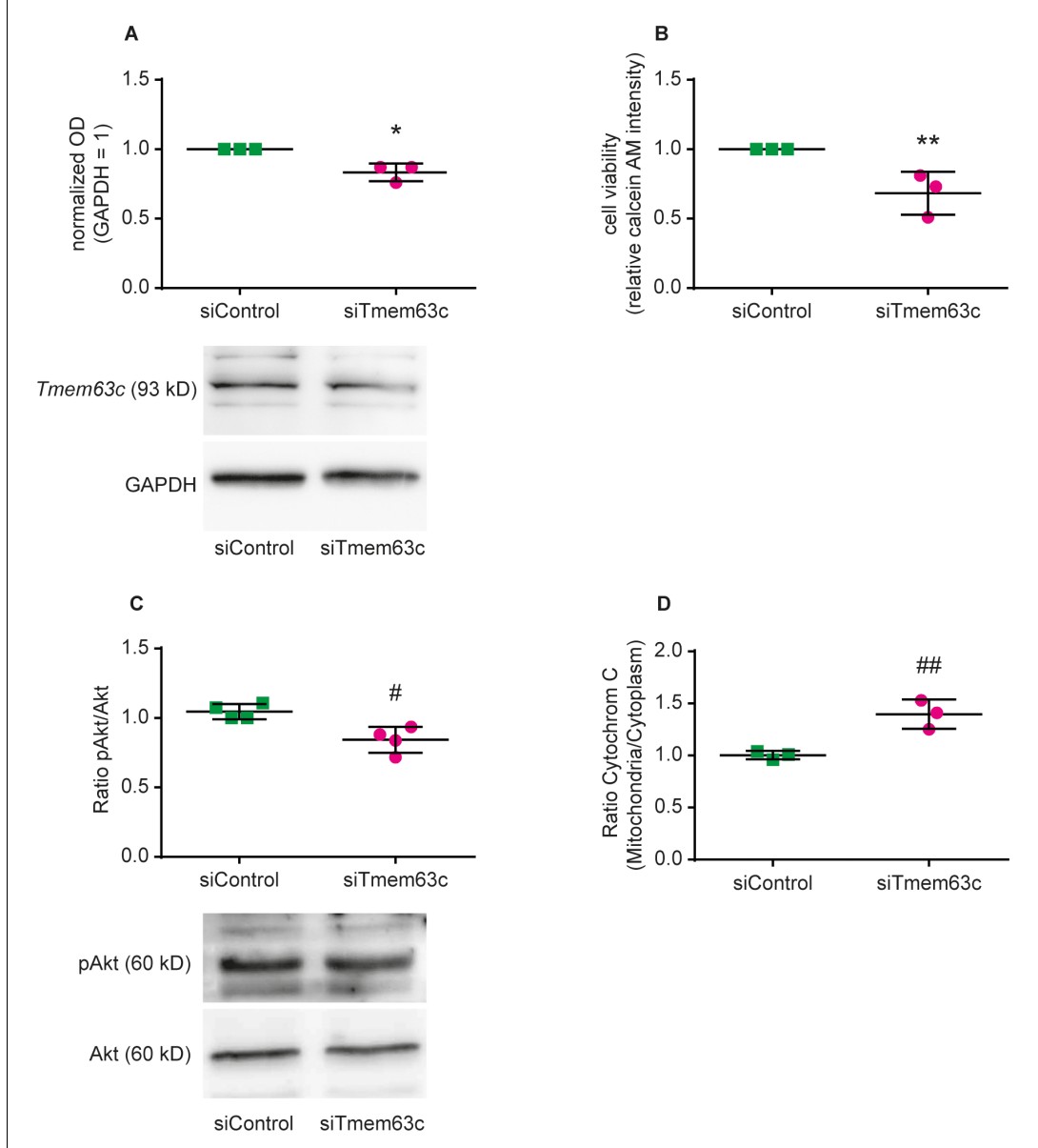

**Figure 6.** Functional role of TMEM63C in human podocytes (hPC) in vitro. Shown is the impact of siRNA-mediated inhibition of TMEM63C (siTMEM63C) or treatment of hPC with a corresponding nonsense negative control (siControl). (**A**) TMEM63C protein expression normalized against GAPDH as a loading control (*p=0.011). Shown is a representative Western blot. (**B**) Cell viability determined via measurement of calcein acetoxymethyl (AM) fluorescence intensity in hPC (**p=0.024). (**C**) Phosphorylation state of protein kinase B (pAKT) normalized against the expression of total AKT (AKT) (#p=0.0094). Shown is a representative Western blot, GAPDH is used as a loading control. (**D**) Pro-apoptotic cytochrome C transition from mitochondria to the cytoplasm of hPC (##p=0.0096). The mean ±SD of at least three independent experiments is shown, respectively. Two-tailed Student´s *t*-test was performed for all experiments.

DOI: https://doi.org/10.7554/eLife.42068.020

The following source data is available for figure 6:

**Source data 1.** TMEM63C expression in human podocytes using small interfering RNA (siRNA) methodology.
DOI: https://doi.org/10.7554/eLife.42068.021

mimicking an albuminuria-like phenotype (*Figure 7A*). To reduce *tmem63c* levels in developing zebrafish embryos, we used the morpholino knockdown technology as well as CRISPR/Cas9-mediated somatic mutagenesis (*Bassett et al., 2013*; *Burger et al., 2016*) (*Figure 7B*). In both morpholino-injected embryos and crispants (CRISPR/Cas9-mediated somatic mutants), loss of *tmem63c* did

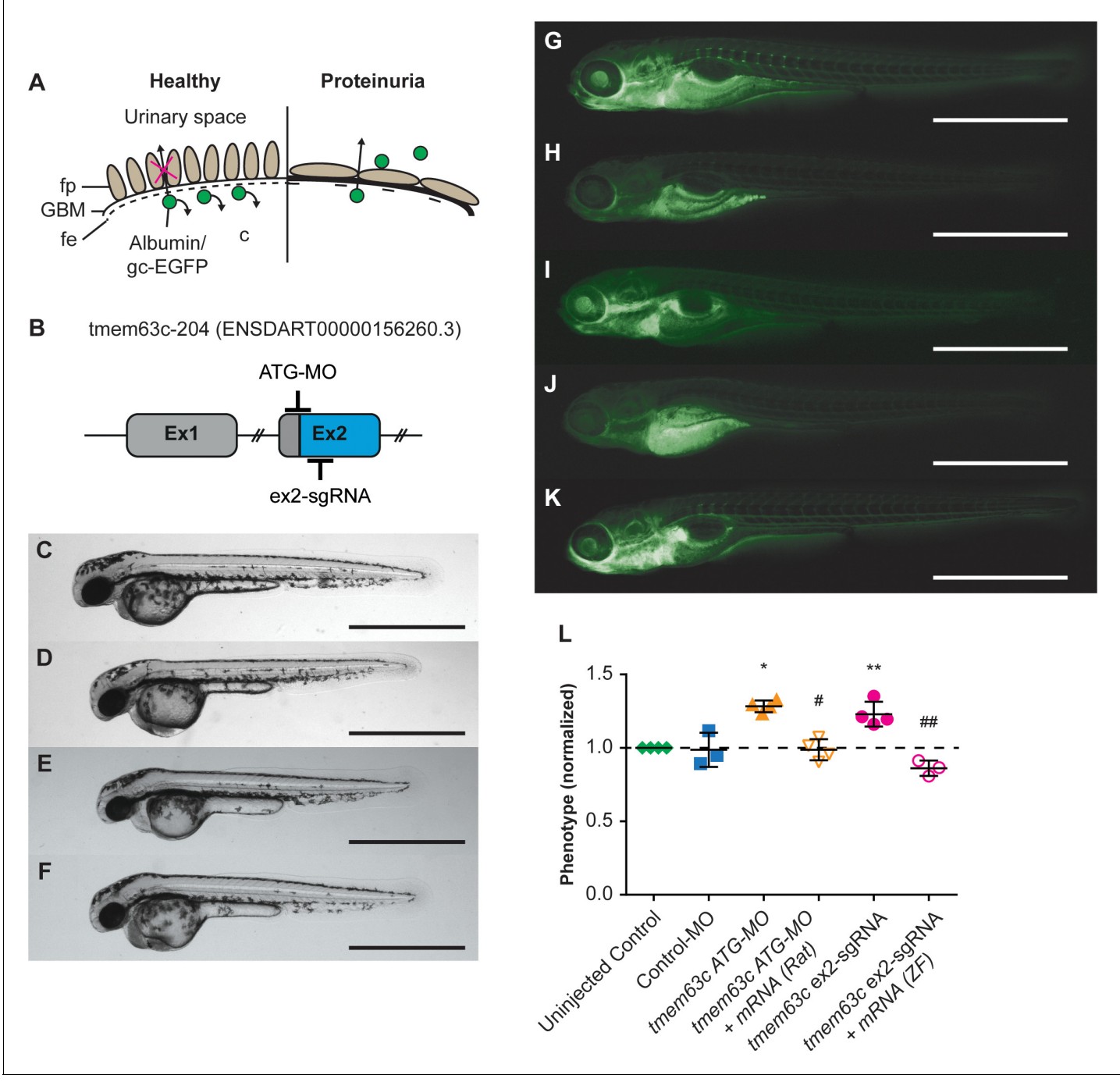

**Figure 7.** Functional assessment of the glomerular filtration barrier (GFB) after loss of transmembrane protein 63c (*tmem63c*) in zebrafish. (**A**) Scheme of the GFB in healthy and albuminuric zebrafish embryos. Green fluorescent protein (gc-EGFP) represents an albumin surrogate and is filtrated after impairment of the GFB. fe, fenestrated endothelium; fp, foot processes of podocytes; GBM, glomerular basement membrane. (**B**) Schematic of *tmem63c* showing the target regions in Exon 2 (ex2) used for Morpholino (MO)- and CRISPR/Cas9-mediated somatic mutagenesis. (**C–F**) Bright field view of wildtype embryos at 48 hr post-fertilization (hpf) in uninjected controls (**C**), ATG-MO injected (**D**), 159.6 ng/µl ex2-sgRNA injected (**E**), and 250 ng/µl ex2-sgRNA injected embryos (**F**). Scale bar = 1 mm. (**G–K**) Fluorescence microscopy of *Tg[fabp10a:gc-EGFP]* embryos at 120 hpf. Uninjected control with clearly visible gc-EGFP fluorescence in the trunk vasculature ('fluorescent') (**G**). *tmem63c* ATG-MO-injected embryo with partial or a complete loss of trunk fluorescence ('deficient-fluorescent') (**H**) and *tmem63c* ATG-MO +*Tmem63*c mRNA (Rat) co-injected embryo (**I**) showing rescue of the phenotype. *Tmem63c* ex2-sgRNA-injected embryo with partial or a complete loss of trunk fluorescence ('deficient-fluorescent') (**J**) and *tmem63c* ex2-sgRNA + *tmem63c* mRNA (ZF) co-injected embryo (**K**) showing rescue of the phenotype. Scale bar = 1 mm. (**L**) Analysis of gc-EGFP in the trunk vasculature. Shown are embryos categorized as 'deficient-fluorescent' (df), see Materials and method section and ***Figure 7—figure supplement 2*** for details. Experimental groups are normalized to the corresponding uninjected control group per experiment. Uninjected Control (*n* = 1198); Control-MO

*Figure 7 continued on next page*

*Figure 7 continued*

(*n* = 189); *tmem63c* ATG-MO (*n* = 227); tmem63c ATG-MO +Tmem63c mRNA (Rat) (*n* = 230); *tmem63c* ex2-sgRNA (*n* = 371); *tmem63c* ex2-sgRNA + *tmem63c* mRNA (ZF) (*n* = 126); One-way ANOVA with Bonferroni's multiple comparisons test. Values plottet: mean ±SD, dashed line at y = 1 indicates the uninjected control level; *p=0.0002 vs. uninjected Control, #p<0.0001 vs. *tmem63c* ATG-MO. **p=0.0014 vs. uninjected Control, ##p<0.0001 vs. *tmem63c* ex2-sgRNA. Data points in the graph represent the ratio per independent experiment, %(Uninjected Control (df)) / %(experimental group (df)), N ≥ 3. (See *Figure 7—figure supplement 1* and *Figure 7—figure supplement 2*).
DOI: https://doi.org/10.7554/eLife.42068.022

The following source data and figure supplements are available for figure 7:

**Source data 1.** Functional assessment of the GFB after tmem63c-knockdown using tmem63c ex2-sgRNA and ex2-sgRNA and tmem63c ATG-MO.
DOI: https://doi.org/10.7554/eLife.42068.026
**Figure supplement 1.** Functional assessment of the glomerular filtration barrier (GFB) in *Tg[fabp10a:gc-EGFP]* zebrafish embryos.
DOI: https://doi.org/10.7554/eLife.42068.023
**Figure supplement 1—source data 1.** Functional assessment of the GFB after tmem63c-knockdown using tmem63c ex2-sdMO.
DOI: https://doi.org/10.7554/eLife.42068.024
**Figure supplement 2.** Excerpt from sequence alignment of *tmem63c* mRNA zebrafish (NM_001159836) vs *Tmem63c* mRNA rat (NM_001108045.1).
DOI: https://doi.org/10.7554/eLife.42068.025

not result in any visible developmental malformations apart from mild pericardial edema (*Figure 7C–F*). Knockdown of *tmem63c* using morpholino technology resulted in a significant decrease in gc-EGFP fluorescence in the trunk vasculature at 120 hr post fertilization (hpf) (*Figure 7H*). We corroborated this finding in *tmem63c* crispants (*Figure 7J*, *Figure 7—figure supplement 1D*) as well as by using another splice-blocking morpholino (*Figure 7—figure supplement 1E–I*); both experiments showed a similar albuminuria-like phenotype. To verify the specificity of the observed phenotype rescue experiments were carried out by co-injection of zebrafish *tmem63c* mRNA with *tmem63c* sgRNA/Cas9 complexes and rat *Tmem63c* mRNA (mRNA sequence identity vs. zebrafish = 65.82% (Clustal 2.1), *Figure 7—figure supplement 2*) with *tmem63c* ATG-MO, respectively. For both cases similarly, the albuminuria-like phenotype could be specifically rescued proving knockdown specificity on the one hand and functional conservation of *tmem63c* across species on the other hand (*Figure 7I,K and L*). Our data indicate that *tmem63c* may regulate the GFB integrity.

To understand the possible functional changes in GFB in more detail, we deployed electron microscopy to visualize the GFB ultrastructure in embryos with reduced *tmem63c* levels. We observed significant changes in podocyte foot process morphology manifested by foot process effacement (*Figure 8A–C*). Quantitative analysis in *tmem63c* crispants revealed a significant increase in the foot process width compared to uninjected controls and Cas9-controls (*Figure 8D*) with concomitant significant decrease of the number of slit diaphragms per μm glomerular basement membrane (GBM) (*Figure 8E*). To analyze, whether the observed albuminuria-like phenotype upon *tmem63c*-deficiency is associated with the loss of podocytes, we utilized confocal microscopy to image glomeruli of *Tg(wt1b:EGFP)* embryos. In this analysis, *tmem63c* crispants (*tmem63c* ex2-sgRNA) showed a widened Bowman´s space and increased glomerular volumes compared to uninjected controls (133457 ± 59547 μm$^3$ vs 67067 ± 21933 μm$^3$, p=0.04) as quantified by 3D surface reconstruction. In addition, we observed dilated capillary loops in the crispants (*Figure 8F–H*). Quantification of podocyte cell number in embryos with reduced *tmem63c* levels revealed no changes in absolute cell number (*Figure 8I*, *Videos 1–3*), while relative podocyte cell number normalized to the total glomerular volume was significantly decreased compared to uninjected controls (*Figure 8J*, *Videos 1–3*). Collectively, our data indicate the conserved role for *tmem63c* in GFB function between fish, rodents, and humans.

## Discussion

Previous genetic analysis in the fawn-hooded hypertensive (FHH) rat model led to the identification of naturally occurring genetic variants in RAB38, member RAS oncogene family (*Rab38*), and shroom family member 3 (*Shroom3*) that successfully complemented studies in humans supporting their role for albuminuria (*Rangel-Filho et al., 2013*; *Yeo et al., 2015*). Interestingly, the genetic variant in *Rab38* was linked to altered tubular (*Rangel-Filho et al., 2013*; *Teumer et al., 2016*), and the variant

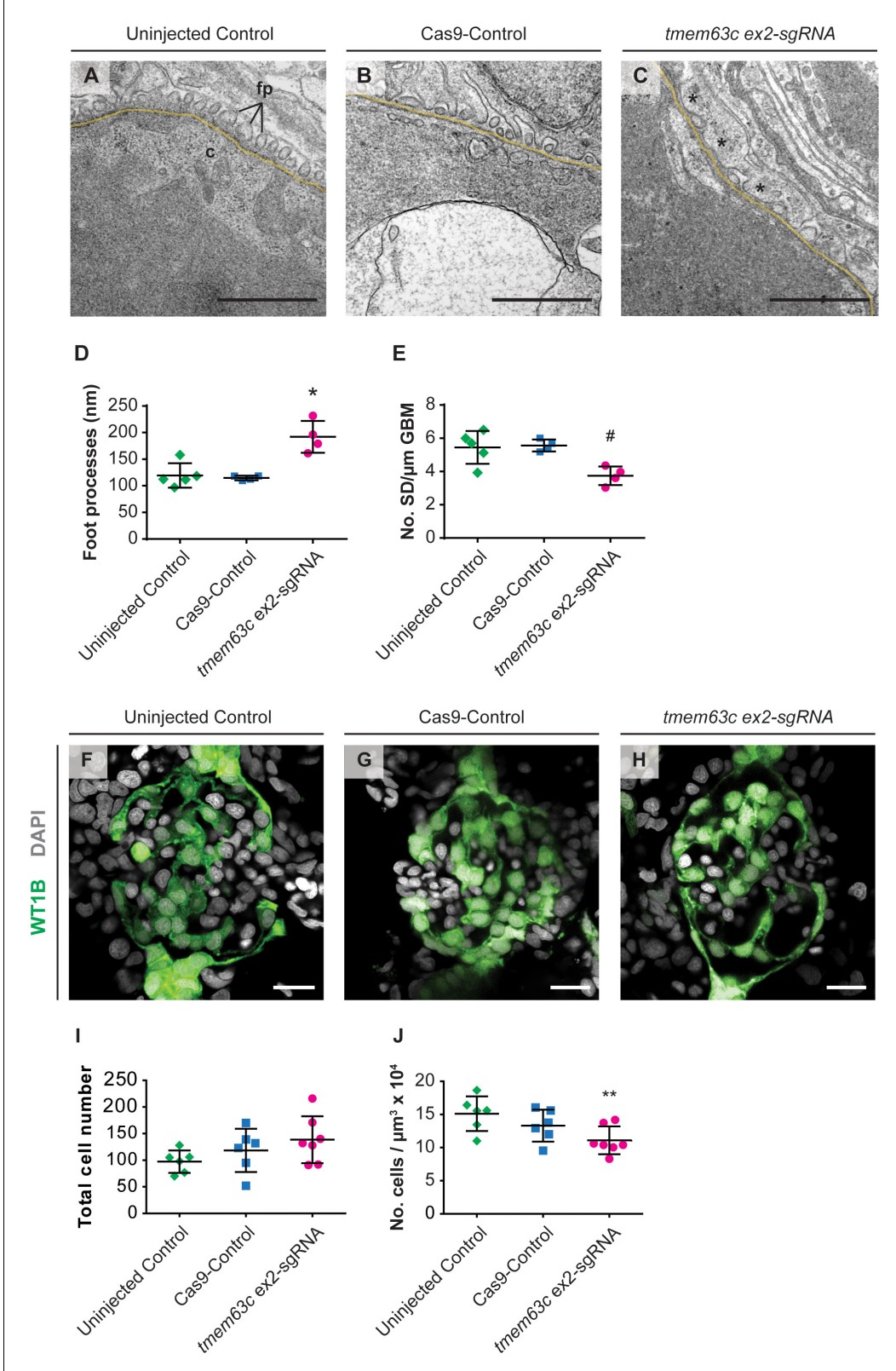

**Figure 8.** Ultrastructural and morphological analysis of glomerular structures after loss of *tmem63c* in zebrafish. (**A–E**) Electron microscopy and quantitative assessment of GFB ultrastructure. Representative electron microscopy pictures of the GFB in uninjected Controls (**A**), Cas9-Controls (**B**) and after *tmem63c* knockdown (**C**), asterisks indicate effaced podocyte foot processes). Quantitative analysis of podocyte foot process width (**D**) and number of slit diaphragms per μm GBM (**E**). Uninjected Control (*n* = 5); Cas9-Control (*n* = 4); *tmem63c* ex2-sgRNA (*n* = 4); Scale bar = 1 μm; values

*Figure 8 continued on next page*

*Figure 8 continued*

plottet: mean ±SD; One-way ANOVA with Bonferroni's multiple comparisons test; *p=0.0019 vs. uninjected Control, *p=0.0017 vs Cas9-Control; # p=0.0171 vs. uninjected Control, # p=0.0148 vs. Cas9-Control. (F–J) Confocal microscopy and analysis of absolute and relative podocyte cell number in *Tg(wt1b:EGFP)* at 96 hpf. Representative confocal microscopy pictures of glomeruli in uninjected Controls (F), Cas9-Controls (G) and after *tmem63c* knockdown (H). Quantitative analysis of absolute (I) and relative (J) podocyte cell number. Relative podocyte cell number has been obtained after normalization to the glomerular volume. Uninjected Control (*n* = 6); Cas9-Control (*n* = 6); tmem63c ex2-sgRNA (*n* = 7); Scale bar = 15 µm; values plottet: mean ±SD; One-way ANOVA with Bonferroni's multiple comparisons test; **p=0.0421 vs. uninjected Control.

DOI: https://doi.org/10.7554/eLife.42068.027

The following source data is available for figure 8:

**Source data 1.** Tables and legends.
DOI: https://doi.org/10.7554/eLife.42068.028

in *Shroom3* to altered glomerular (*Yeo et al., 2015*) albumin handling. Thus, animal models have been proven useful not only for the explanation of missing heritability (*Chatterjee et al., 2013*), but also for the elucidation of the differences between glomerular and tubular origins of albuminuria (*Rangel-Filho et al., 2013*; *Yeo et al., 2015*). Currently, in GWAS meta-analyses in general population cohorts, only cubilin (*CUBN*) has been significantly associated with albuminuria (*Böger et al., 2011*). In addition, variants in *HS6ST1* and near *RAB38/CTSC* were implicated in albuminuria in patients with diabetes (*Teumer et al., 2016*). Nevertheless, in parallel to the findings obtained in BP GWAS (*Hoffmann et al., 2017*; *Warren et al., 2017*), only a small fraction of the estimated heritability of albuminuria can be attributed to the identified genes (*Langefeld et al., 2004*; *Teumer et al., 2016*). The MWF rat, in which we identified at least 11 albuminuria QTL, highlights the polygenic nature of complex traits such as kidney damage with albuminuria in hypertension (*Schulz and Kreutz, 2012*). Here, we focused on a major albuminuria QTL (*Schulz and Kreutz, 2012*; *Schulz et al., 2003*) and refined the candidate region to a sub-QTL comprising 4.9 Mb. We confirmed that both the albuminuria and the nephron deficit phenotype map to the same genomic region supporting a genetic link between albuminuria and embryonic/fetal nephron development (*Wang and Garrett, 2017*). In addition, the informative congenic lines with differential albuminuria development showed similar BP values by which we showed that this genomic region affects albuminuria development independently from BP changes. In the targeted NGS analysis, multiple non-deleterious variants but no clear candidate was identified. However, significant signs of selective sweeps in this region, potentially leading to an enrichment of multiple non-deleterious alleles due to genetic hitchhiking, were detected. This is in agreement with a recent systematic genome sequencing study of laboratory inbred rat strains indicating that private single-nucleotide variants are highly concentrated in a small number of discrete regions of the genome (*Atanur et al., 2013*). The authors of this report hypothesized that variants that are unique to a single strain reside within these regions because many of these regions were positively selected in the initial phenotype-driven derivation of these strains. Surprising - and challenging however - remains our observation that no obvious causative variant for the kidney damage phenotype was observed in our targeted NGS analysis. Notwithstanding, complex traits

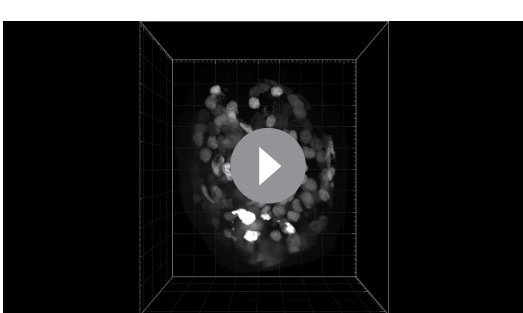

**Video 1.** Representative spot segmentation of podocyte nuclei and 3D surface reconstruction of glomeruli in uninjected controls of *Tg(wt1b:EGFP)* zebrafish embryos. The videos show a maximum intensity projection of the masked DAPI channel acquired by confocal microscopy of *Tg(wt1b:EGFP)* zebrafish embryos at 96 hpf. The DAPI channel of DAPI⁺/EGFP⁺-cells visualized here represents podocyte nuclei. Blue spots show the podocyte nuclei identified by software-based spot segmentation. Spots were counted for quantification of the absolute podocyte cell number. The grey surface represents the 3D surface reconstruction of glomeruli containing all DAPI⁺/EGFP⁺ nuclei, which was used for quantification of the total glomerular volume. Shown here are representative analysis results of uninjected controls (*Video 1*), Cas9-Controls (*Video 2*) and *tmem63c* ex2-sgRNA-injected embryos (*Video 3*).

DOI: https://doi.org/10.7554/eLife.42068.029

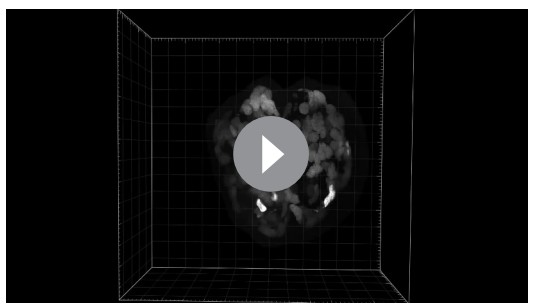

**Video 2.** Representative spot segmentation of podocyte nuclei and 3D surface reconstruction of glomeruli in Cas9 - injected controls (Cas9 - controls) of Tg(wt1b:EGFP) zebrafish embryos.
DOI: https://doi.org/10.7554/eLife.42068.030

**Video 3.** Representative spot segmentation of podocyte nuclei and 3D surface reconstruction of glomeruli in *tmem63c* ex2-sgRNA - injected Tg(wt1b: EGFP) zebrafish embryos
DOI: https://doi.org/10.7554/eLife.42068.031

can also be a consequence of gene expression changes resulting in dysregulation of physiological pathways, a concept which becomes increasingly recognized as RNA sequencing is employed to distinguish tissue-specific gene expression patterns in a variety of complex diseases (*Joehanes et al., 2017*; *Kirsten et al., 2015*).

We therefore complemented the DNA re-sequencing by compartment-specific RNA-Seq analysis which is rational, because MWF rats develop early glomerular changes that precede albuminuria development and show early podocyte injury in parallel with the onset of albuminuria (*Ijpelaar et al., 2008*). We identified *Tmem63c* as a positional candidate based on its mRNA expression pattern with differential glomerular expression in allele-specific rat models. Immunohistochemistry analysis in MWF kidneys identified TMEM63C protein in podocytes, and in contrast to the observed marked glomerular mRNA upregulation only a modest downregulation. Bioinformatics analysis of our NGS and RNA-Seq data has not yet provided a potential explanation for this discrepancy, for example interstrain differences related to exon usage or other events shown to mediate protein expression control on the post-transcriptional level. The relationship between protein levels and their coding transcripts is, however, rather complex and factors such as spatial and temporal variations of mRNAs, as well as the local availability of resources for protein biosynthesis, strongly influence this relationship (*Liu et al., 2016*). Spatial variation in expression regulation during renal injury development could be specifically important in podocytes due to their unique ultrastructural and molecular anatomy (*Endlich et al., 2017*). To explore the potential clinical relevance of TMEM63C for kidney damage, we selected patients with FSGS because they mirror the disease pattern observed in the MWF model (*D'Agati et al., 2011*; *Lim et al., 2016*; *Yu et al., 2016*). Importantly, patients with FSGS exhibit a significant decrease of TMEM63C protein levels in podocytes with a global loss of glomerular expression in the majority of patients. Furthermore, the loss of TMEM63C was associated with a decrease and altered granular staining pattern of nephrin protein in glomeruli of FSGS patients as previously reported (*Doublier et al., 2001*; *Wernerson et al., 2003*). The latter has been previously shown to correspond to the degree of foot process effacement (*Wernerson et al., 2003*). While our data cannot establish a functional link between changes in nephrin and TMEM63c in FSGS, they nonetheless demonstrate a concomitant deficiency of both proteins in this setting.

To further validate *Tmem63c*, we tested its functional relevance for albuminuria development in zebrafish (*Danio rerio*). Our data confirm that in both *tmem63c* morphants and crispants, loss of *tmem63c* leads to a phenotype indicative of a GFB defect. Importantly, rescue of the observed phenotypes in zebrafish with rat *Tmem63c* mRNA pointed to the functional conservation across species. Ultrastructural analysis in zebrafish embryos demonstrated the role of *tmem63c* for the GFB integrity by revealing overt effacement of podocyte foot processes upon gene deficiency. Further structural analysis by confocal microscopy revealed dilated capillary loops and enlarged glomerular volumes with a reduction of relative podocyte number in relation to glomerular volume (podocyte density) in *tmem63c*-deficient embryos. Thus, our data reveal an important role of *tmem63c* for normal development of capillary structure and podocyte function of the glomerulus in zebrafish.

*Tmem63c* belongs to the TMEM (transmembrane protein) gene family comprising more than 300 different proteins with about 580 transcript variants (*Wrzesiński et al., 2015*). These proteins are predicted components of cellular membranes. However, the function of the majority of TMEM proteins - including *Tmem63c* - is currently unclear. There are no previous reports that demonstrate TMEM63C expression in the kidney, while assessment of the human protein atlas database indicates TMEM63C expression in human kidney in both the glomerular and tubular compartments (*Lundberg et al., 2010*); 'Tissue expression of TMEM63C - Staining in kidney' - *The Human Protein Atlas, 2019*; *Uhlén et al., 2015*). T*mem234*, another member of the TMEM family, may represent a component of the basal membrane of podocytes and has recently been associated with proteinuria development in zebrafish (*Rodriguez et al., 2015*). The mechanism by which changes in *Tmem63c* expression cause podocyte damage and thus contribute to the final common pathway of injury in FSGS (*De Vriese et al., 2018*) remains however unclear and cannot be elucidated by our current analysis. Accordingly, our data do not allow to describe possible interactions between TMEM63c and the heterogeneous group of known causative FSGS genes. Nevertheless, previous reports suggested that T*mem63c* and two other genes of the tmem63-family, i.e. *Tmem63a* and *Tmem63b,* are mammalian homologues of the hyperosmolarity-activated cation channel proteins AtCSC1 and its paralogue OSCA1 in plants (*Hou et al., 2014*; *Zhao et al., 2016*). Based on these reports, it appeared tempting to speculate that Tmem63c may be involved in mechanosensing in podocytes providing thus a functional basis for interactions with cytoskeleton genes or slit pore proteins, for example nephrin, in podocytes. However, the potential role of Tmem63c as a hyperosmolarity-activated cation channel remains controversial, since the activation by hyperosmolarity was not confirmed in a more recent study (*Murthy et al., 2018*). Moreover, the latter report indicated that *Tmem63c* is phylogenetically divergent from *Tmem63a* and *Tmem63b* and lacks their function to induce stretch-activated ion currents. Consequently, further studies are needed to characterize the functional role of *Tmem63c* and to explore its potential to influence podocyte function as a novel target for therapeutic intervention (*Endlich et al., 2017*; *Forst et al., 2016*; *Wieder and Greka, 2016*). In contrast to our experiments in zebrafish, in the explorative analysis in biopsies of FSGS patients we cannot differentiate between a primary (causative) or secondary effect of TMEM63C expression loss in patients. Nevertheless, taken together with our experimental analysis, the data clearly support *Tmem63c* as a novel candidate for further translational research on kidney damage.

## Materials and methods

**Key resources table**

| Reagent type (species) or resource | Designation | Source or reference | Identifiers | Additional information |
|---|---|---|---|---|
| Antibody | anti-TMEM63C | Perbio Science Germany; this paper | epitope: GLRGFAREL DPAQFQEGLE | 1:1600 for rat tissue, 1:800 for human biopsies |
| Antibody | rabbit anti-WT1 | Santa Cruz | RRID:AB_632611 | 1:500 |
| Antibody | rabbit anti-nephrin | Abcam | RRID:AB_944400 | 1:750 |
| Antibody | Goat anti-rabbit EnVision HRP conjugate | Dako | | |
| Antibody | GAPDH | Calbiochem | | |
| Antibody | AKT | Merck Chemicals GmbH | | |
| Antibody | phospho-AKT Ser$_{473}$ | Merck Chemicals GmbH | | |
| Biological sample (*Homo sapiens*) | Renal biopsy samples of patients with FSGS | archive of the Department of Pathology of the Leiden University Medical Center (LUMC) | | For patient information see Table 4 |

*Continued on next page*

*Continued*

| Reagent type (species) or resource | Designation | Source or reference | Identifiers | Additional information |
|---|---|---|---|---|
| ell line (*Homo sapiens*) | hPC | *Saleem et al., 2002* | RRID:CVCL_W186 | |
| Gene (*Danio rerio*) | *tmem63c* | NA | ZFIN: ZDB-GENE-120928–2 | |
| Gene (*Homo sapiens*) | TMEM63C | NA | Ensembl: ENST00000298351.4 | |
| Gene (*Rattus norvegicus*) | Tmem63c | NA | Ensembl: ENSRNOT00 000015571.6 | |
| Other | DAPI stain | Sigma Aldrich | | stock solution 1 mg/ml diluted 1:2000 in PBS |
| Other | DNA-Seq database | this paper | GEO and SRA: Submission ID: SUB2950675 and BioProject ID: PRJNA398197 | See Data availability |
| Other | RNA-Seq database | this paper | GEO and SRA: accession GSE102546 | See Data availability |
| Recombinant DNA reagent | *tmem63c* (cDNA) zebrafish | this paper | | Infusion cloning primer sequences for cDNA synthesis: forward: GCTTGATATC GAATTCATG GCGTTTGAGTCCT GGCCTGC; reverse: CGGGCTGCAGGA ATTCTCACTGAA AAGCCACCGGACTG; Sequence additional primer for amplification of ORF: GTGCAGAAAC TAATGAAGCTGG; Progenitors: *tmem63c* (cDNA); pBluescript II SK(+) |
| Sequence-based reagent | *tmem63c* ATG-MO | Gene Tools LLC Philomath | | sequence: 5'-CAGGCCAGGAC TCAAACGCCATTGC-3' |
| Sequence-based reagent | *tmem63c* ex2-sdMO | Gene Tools LLC Philomath | | sequence: 5'-TGTTATCATAGATGA TGTACCAGCC-3' |
| Sequence-based reagent | *tmem63c* ex2-sgRNA | this paper | | Sequence synthesis forward primer with CRISPR target site underlined: GAAATTAATACGACT CACTATAGGACGTC AGGAGTTTCCTGAGTT TTAGAGCTAGAAATAGC |
| Sequence-based reagent | *tmem63c* ex2-sgRNA primers flanking CRISPR target site | BioTez Berlin-Buch GmbH | | Sequences: forward: CAAATGGTGAAC ACTTGTGAATC, reverse: CTGCGGTT TACTGCGGAGATG |
| Sequence-based reagent | siTMEM63C | Sigma-Aldrich Chemie GmbH | | |
| Strain, strain background (*Danio rerio*) | Tg(fabp10a:gc-EGFP) | *Zhou and Hildebrandt, 2012* | | |

*Continued on next page*

*Continued*

| Reagent type (species) or resource | Designation | Source or reference | Identifiers | Additional information |
|---|---|---|---|---|
| Strain, strain background (*Danio rerio*) | *Tg(wt1b:GFP)* | *Perner et al., 2007* | | |
| Strain, strain background (*Rattus norvegicus*) | MWF/Rkb | Own colony Charité – Universitätsmedizin Berlin, Germany | http://dels.nas.edu/ilar/ (laboratory code Rkb); *Schulz and Kreutz, 2012*; RRID:RGD_724569 | |
| Strain, strain background (*Rattus norvegicus*) | SHR/Rkb | Own colony Charité – Universitätsmedizin Berlin, Germany | http://dels.nas.edu/ilar/ (laboratory code Rkb); *Schulz and Kreutz, 2012*; RRID:RGD_631696 | |
| Strain, strain background (*Rattus norvegicus*) | MWF-6<sup>SHR</sup> | Own colony Charité – Universitätsmedizin Berlin, Germany | http://dels.nas.edu/ilar/ (laboratory code Rkb); *Schulz and Kreutz, 2012*; RRID:RGD_1641831 | |
| Strain, strain background (*Rattus norvegicus*) | Congenic strains see *Figure 1* | Own colony Charité – Universitätsmedizin Berlin, Germany; this paper | | For generation of congenic strains see Materials and method section |

## Rat animals

Male rats were obtained from our MWF/Rkb (RRID:RGD_724569, laboratory code Rkb, http://dels.nas.edu/ilar/) and SHR/Rkb (RRID:RGD_631696, laboratory code Rkb, http://dels.nas.edu/ilar/) colonies at the Charité – Universitätsmedizin Berlin, Germany. The consomic MWF-6$^{SHR}$ (RRID:RGD_1641831) was previously described (*Schulz et al., 2007*). Rats were grouped under conditions of regular 12 hr diurnal cycles with an automated light switching device and climate-controlled conditions at a room temperature of 22°C. The rats were fed a normal diet containing 0.2% NaCl and had free access to food and water.

A panel of eight congenic rat lines MWF.SHR-(*D6Rat1-D6Rat30*), MWF.SHR-(*D6Rat1-D6Rat106*), MWF.SHR-(*D6Rat1-D6Mit8*), MWF.SHR-(*D6Rat1-D6Rat121*), MWF.SHR-(*D6Rat1-D6Mgh4*), MWF.SHR-(*D6Rat1-D6Rat81*), MWF.SHR-(*D6Rat1-D6Rat115*), and MWF.SHR-(*D6Rat1-D6Rat184*) was generated by transfer of different nested SHR segments onto the MWF background. For this procedure, male and female rats of the MWF-6$^{SHR}$ breeding, that were homozygous for all MWF chromosomes except RNO6 and heterozygous for RNO6, were intercrossed (*Schulz et al., 2003*). All experimental work in rat models was performed in accordance with the guidelines of the Charité-Universitätsmedizin Berlin and the local authority for animal protection (Landesamt für Gesundheit und Soziales, Berlin, Germany) for the use of laboratory animals. The registration numbers for the rat experiments are G 0255/09 and T 0189/02.

## Determination of albuminuria, direct BP and glomerular density in rats

Urinary albumin excretion was measured as reported (*Kreutz et al., 2000*). Direct intra-arterial BP measurements were performed in awake male rats at 14 weeks of age as previously described (*Kreutz et al., 1995*; *Schulz et al., 2010*). For determination of glomerular density, animals were sacrificed under ketamine-xylazine anesthesia (87 and 13 mg/kg body wt, respectively) at week 4. The right kidney was fixed in methacarn and embedded in paraffin. Tissue samples were cut into 5-μm-thick histological sections (*Figure 1G–I*) and stained with the periodic acid-Schiff (PAS) technique. Section analysis was performed by a photomicroscope Axiophot (Zeiss) and a digital camera system AxioCam MRc Rev. 3 FireWire (Zeiss) at a 10x magnification. Glomerular density was calculated using the formula n = G/FA(D + T) as reported (*ELIAS et al., 1961*; *Lucas et al., 1997*). Glomeruli in 20–25 fields for each sample were counted in the outer cortex zone (*Figure 1H,I*). Glomerular diameter was calculated by the AxioVision release 4.8.2 software program (Zeiss). This method was validated by comparison with the absolute nephron numbers as determined by the physical fractionator method in rat strains as previously reported (*Gundersen, 1986*; *Schulz et al., 2007*).

## NGS of the candidate region on RNO6

Based on the fine mapping results of the congenic MWF strains, a target candidate region of about 5.63 Mbp (chr6:105.8–111.43 Mb, R. norvegicus, ENSEMBL rn6.0) (*Yates et al., 2016*) was defined for subsequent next-generation resequencing analysis. The solution-based SureSelectXT (Agilent Technologies) capture method was applied for custom target enrichment of the defined region according to the manufacturer's instructions starting with 3 µg genomic rat DNA of MWF, SHR, MWF-6$^{SHR}$, MWF.SHR-(*D6Rat1-D6Mgh4*), and MWF.SHR-(*D6Rat1-D6Rat81*) (*n* = 3, each), uniquely labelled by index tags.

Library quality control and final quantification for subsequent pooling of the 16 sequencing libraries was performed using the 2100 Bioanalyzer instrument (Agilent Technologies). The pooled library was paired end sequenced (2 × 76 cycles plus index read) on a MiSeq system (Illumina) using the MiSeq reagent kit v3 (Illumina). The CASAVA software package v1.8.2 (Illumina) was used for demultiplexing of the sequencing reads and conversion to fastq data for further analysis. The resulting high-quality reads for identification of SNPs and short INDELs were mapped to the reference genome according to the annotation release Rat Genome Sequencing Consortium (RGSC) genome assembly v6.0 using BWA software, version 0.6.2 (*Li and Durbin, 2009*). Mapped reads were processed and calling of SNPs and short INDELs were performed by the GATK pipeline, version 2.8 (*Van der Auwera et al., 2013*). Effects of the genomic variations were evaluated with the SnpEff software tool, version 3.3 (*Cingolani et al., 2012*). Impairment of protein function by common exonic variants in MWF and SHR were analysed using the PROVEAN algorithm (*Choi et al., 2012*). A PROVEAN Score <2.5 was considered significant for genes.

The Tajima's D statistic (*Tajima, 1989*) was used to test for signatures of selection in the region of interest utilizing the vcftools software (Vs. 0.1.13) (*Danecek et al., 2011*).

## Rat glomeruli isolation

Different protocols were used for isolation of glomeruli from male rats at 4 and 8 weeks of age, due to the different body size. Rats were anesthetized with ketamine-xylazine (87 and 13 mg/kg body weight, respectively). In 4-week-old rats, the abdominal artery was catheterized and kidneys were perfused with 10 ml 1x phosphate buffered saline (PBS) and subsequently with 20 ml ferrous solution (12.5 g ferric oxide (Iron(II/III) powder <5 micron, 98%; Sigma- Aldrich Chemie GmbH) suspended in 1000 ml 1x PBS. Kidneys were removed, decapsulated and passed through a 125 µm steel sieve (Retsch GmbH) with 1x PBS. The glomeruli containing ferrous particles were gathered by a magnet, snap-frozen and stored at −80°C. Kidneys of 8-week-old rats were removed, decapsulated and passed through a 125 µm steel sieve with 1x PBS. The filtrate was put on a 71 µm steel sieve (Retsch GmbH) to separate glomeruli from the flow-through. Glomeruli were washed off the sieve with 1x PBS, centrifuged, immediately snap-frozen and stored at −80°C.

## Transcriptome analysis

RNA sequencing (RNA-Seq) was performed in glomerular RNA of male MWF and SHR rats at week 4 (*n* = 3, each). The NEBNext Poly(A) mRNA magnetic isolation module followed by library preparation using NEBNext Ultra RNA Library Prep Kit for Illumina (New England BioLabs) was applied on 1 µg total RNA to generate a cDNA library for subsequent paired end (80 cycles) sequencing on the NextSeq 500 system (Illumina) using v2 chemistry yielding in about 415M single reads. RNA and library quality control was performed using the Bioanalyzer RNA 600 Nano and High-Sensitivity DNA Analysis Kit (Agilent Technologies), respectively. The KAPA Library Quantification Kit (Kapa Biosystems) was used for library quantification.

Initial quality control of the raw data was performed using Cutadapt version 1.9 (*Martin, 2012*) program. Raw reads were quality trimmed (minimal base quality: 25, minimal read length after trimming: 70 nt), adapter sequences were removed from reads 3' ends. TopHat2 version 2.1.0 (*Trapnell et al., 2009*; *Trapnell et al., 2010*) software tool together with Bowtie2 aligner version 2.2.3 (*Langmead and Salzberg, 2012*) was used for read mapping against ENSEMBL rn6.0 reference assembly (*Yates et al., 2016*). After the reads have been mapped to the reference genome, the Cufflinks version 2.2.1 (*Trapnell et al., 2010*) program together with Ensembl (release 81) gene annotation (*Aken et al., 2016*), baw093) were used to assemble transcripts and estimate their abundances. Differential expression analysis was performed using both Cuffdiff version 2.2.1 software package

(*Trapnell et al., 2010*) and DESeq2 R package version 1.12.4 (*Love et al., 2014*). Genes having absolute fold change value <1.5 were excluded from further analysis. Genes were considered significantly differentially expressed if the corresponding adjusted p-value was less than 0.05.

## Reverse transcription and qPCR

First-strand cDNA synthesis was carried out on 2 µg of total RNA using the First Strand cDNA Synthesis Kit (Fermentas Life Sciences) following the manufacturer's protocol. Isolated glomeruli preparations of rat strains were analyzed at week 4 and week 8. qPCR of each gene was performed in a 7000 Real-Time PCR System (Applied Biosystems) with version 1.2.3 software or a 7500 Fast Real-Time PCR System with version 2.0.6 software (Applied Biosystems) using the comparative quantification cycle method as reported (Fast SYBRGreen Master Mix or Power SYBR Green PCR Master Mix; Applied Biosystems) (*Schulz et al., 2008*). Primers are listed in *Figure 4—source data 2*. Normalization of expression data was done by the reference gene hydroxymethylbilane synthase (*Hmbs*) (*Schulz et al., 2008*). For all analyses, three technical replicates of each animal/experiment were performed. Genes with low mRNA expression levels were only considered when the quantification cycles (Cqs) were ≥30 and the Cqs of the no-template controls were at least 5 Cqs delayed. Acyl-CoA thioesterase 3 (Acot3) demonstrated low expression levels and was therefore not analyzed.

## Immunohistochemistry of TMEM63C in rat and human kidneys

For determination of protein expression of TMEM63C an anti-TMEM63C antibody (epitope: G LRGFARELDPAQFQEGLE, custom antibody production: Perbio Science Germany) was generated. The epitope does not cross react with TMEM63A or TMEM63B or other genes. For Wilms tumor 1 (WT1) protein expression analysis, we used a rabbit anti-WT1 antibody (Santa Cruz). For nephrin protein expression analysis, we used a rabbit anti-nephrin antibody (Abcam). Paraffin embedded rat kidney sections and human biopsy samples were cut at 4 µm and incubated with the anti-TMEM63C antibody (1:1600 for rat tissue, 1:800 for human biopsies), the anti-WT1 antibody (1:500) or the anti-nephrin antibody (1:750). Rabbit IgG negative control fraction was used as a negative control in the same concentration as the primary antibody. Goat anti-rabbit EnVision HRP conjugate (Dako) was used as secondary antibody. The staining was visualized using diaminobenzidine as the chromogen and counterstained with haematoxylin.

## Evaluation of TMEM63C staining in MWF and SHR rats

Consecutive slides of MWF and SHR kidney sections stained for TMEM63C and WT1 were evaluated to determine TMEM63C co-localization with podocytes. TMEM63C protein level in glomeruli was analyzed using ImageJ analysis.

**Table 4.** Characteristics of focal segmental glomerulosclerosis patients.

| Patient-specific features | Values[‡] |
|---|---|
| Number of patients | 9 |
| Age, years | 36 ± 23 |
| Sex, male | 5 (55%) |
| SBP, mmHg[*] | 167 ± 38 |
| DBP, mmHg[†] | 106 ± 25 |
| Hypertension | 7 (88%) |
| Proteinuria, g/day[†] | 9.2 ± 4.4 |
| Serum creatinine, µmol/l[†] | 150 ± 43 |
| eGFR, ml/min/1.73 m$^2$[*] | 51 (42–66) |
| Nephrotic syndrome | 8 (89%) |

[*]n = 7; [†]n = 6; [‡] values are reported as number (%), mean ±SD or as median (interquartile range) for eGFR.
DOI: https://doi.org/10.7554/eLife.42068.032

## FSGS patients

Renal biopsy samples of patients with FSGS (*Table 4*) were collected from the archive of the Department of Pathology of the Leiden University Medical Center (LUMC). Demographic data and laboratory data at time of biopsy were retrospectively retrieved from the patients' medical records or pathology reports following the good practice guidelines of the LUMC. All biopsy samples were handled and analyzed anonymously in accordance with the Dutch National Ethics Guidelines (Code for Proper Secondary Use of Human Tissue, Dutch Federation of Medical Scientific Societies). This study is in agreement with the Declaration of Helsinki and the Department of Health and Human Services Belmont Report and the use of the patient biopsies was approved by the medical ethical committee of the LUMC (registration number G16.110). Samples obtained from Eurotransplant donors that were unsuited for transplantation because of technical problems, were used as healthy controls. All sections were scored separately by two observers for TMEM63C intensity as well as for percentage of glomeruli with loss of TMEM63C staining. Each case was given a score for TMEM63C staining intensity: high intensity in >50% of glomeruli, intermediate intensity in >50% of glomeruli, low intensity in >50% of glomeruli or no TMEM63C staining present in >50% of glomeruli. Secondly, sections were scored based on the percentage of glomeruli with loss of TMEM63C expression in podocytes. Per case, the percentage of glomeruli with 1) no loss 2) <25% loss 3) 25–50% loss or 4) >50% loss of TMEM63C expression in podocytes was determined. For nephrin staining analysis, each glomerulus was scored based on staining pattern (linear or granular) and loss of staining (no loss, segmental loss or global loss).

All biopsy samples were handled and analyzed anonymously in accordance with the Dutch National Ethics Guidelines (Code for Proper Secondary Use of Human Tissue, Dutch Federation of Medical Scientific Societies) and in agreement with the Declaration of Helsinki and the Department of Health and Human Services Belmont Report. The use of the patient biopsies was approved by the medical ethical committee of the LUMC.

## Cell lines

Immortalized human podocytes (RRID:CVCL_W186, a kind gift from Professor Moin Saleem, MA, Academic and Children's Renal Unit, University of Bristol, Bristol, UK) (*Saleem et al., 2002*) were used as described (*Eisenreich et al., 2016*). The cell line has previously been authenticated (*Saleem et al., 2002*) and we have confirmed this by expression of podocyte specific markers such as podocin and synaptopodin as recently reported (*Eisenreich et al., 2016*). The cell line tested negative for mycoplasma contamination. Before transfection, human podocytes were starved with FBS-free RPMI 1640 medium overnight. Transfection of cells was performed using 200 nM of TMEM63C-specific siRNAs (siTMEM63C; Sigma-Aldrich Chemie GmbH) or non-sense control siRNAs (siControl; Sigma-Aldrich Chemie GmbH) as well as Lipofectamine 2000 (Life Technologies GmbH). The transfection efficacy of 25% in human podocytes was experimentally determined earlier (*Eisenreich et al., 2016*).

## Western blotting

Western blot analyses were done as described earlier (*Eisenreich et al., 2016*; *Langer et al., 2016*). For detection, specific antibodies against TMEM63C (Thermo Fisher Scientific), GAPDH (Calbiochem), protein kinase B (AKT; Merck Chemicals GmbH), and phospho-AKT (Ser$_{473}$, pAKT; Merck Chemicals GmbH) were used. Quantification of Western blot analyses were done using Gel-Pro Analyzer software version 4.0.00.001 (Media Cybernetics).

## Cytochrome C releasing apoptosis assay

The cytochrome C releasing apoptosis assay kit (BioVision Inc) was used following the manufacturer's protocol as described previously (*Eisenreich et al., 2016*). In brief, $1 \times 10^4$ cells per well were transfected for 48 hr with siTMEM63C or siControl, respectively. After that, cells were lysed and the cytosolic fraction was separated from the mitochondrial fraction. Comparative Western blot analyses of these fractions using a cytochrome C-specific antibody were performed to determine pro-apoptotic translocation of cytochrome C from mitochondria into cytosol.

## Cell viability assay

The calcein AM (acetoxymethyl) cell viability kit (Trevigen Inc) was used as earlier described following the manufacturer's protocol (*Eisenreich et al., 2016*). In brief, $1 \times 10^4$ cells per well were transfected for 48 hr with siTMEM63C or siControl, respectively. Then, human podocytes were washed and incubated with calcein AM working solution for 30 min. Fluorescence was measured at 490 nm excitation and 520 nm emission.

## Zebrafish animals

Zebrafish were bred, raised and maintained in accordance with the guidelines of the Max Delbrück Center for Molecular Medicine and the local authority for animal protection (Landesamt für Gesundheit und Soziales, Berlin, Germany) for the use of laboratory animals, and followed the 'Principles of Laboratory Animal Care' (NIH publication no. 86–23, revised 1985) as well as the current version of German Law on the Protection of Animals.

## Zebrafish morpholino and single guide RNA (sgRNA) microinjections

Injection droplets of approximately 1 nl were injected into one-cell stage zygotes of the zebrafish wild type hybrid strain AB/Tülf and the transgenic lines *Tg(fabp10a:gc-EGFP)* (*Zhou and Hildebrandt, 2012*) and *Tg(wt1b:GFP)* (*Bollig et al., 2009*; *Perner et al., 2007*). Morpholinos (MO) of the following sequences were synthesized by Gene Tools LLC Philomath: *tmem63c* ATG-MO 5'-CAGGC-CAGGACTCAAACGCCATTGC-3', *tmem63c* ex2-sdMO 5'-TGTTATCATAGATGATGTACCAGCC-3', and standard control oligo (Control-MO) 5'-CCTCTTACCTCAGTTACAATTTATA-3'. *Tmem63c* ATG-MO was used at a final concentration of 0.3 mM (*Figure 7B*), *tmem63c* ex2-sdMO was used at a final concentration of 0.5 mM (*Figure 7—figure supplement 1*). sgRNA targeting exon 2 (*Figure 7B*) was generated as described (*Bassett et al., 2013*; *Burger et al., 2016*) using the ex2-sgRNA forward primer with CRISPR target site underlined: GAAATTAATACGACTCACTATA<u>GGACGTCAGGAGTTTCCTGA</u>GTTTTAGAGCTAGAAATAGC and the invariant reverse primer: AAAAGCACCGACTCGGTGCCACTTTTTCAAGTTGATAACGGACTAGCCTTATTTTAACTTGCTATTTCTAGCTCTAAAAC. PCR product was purified with GeneJET Gel Extraction Kit (Thermo Fisher Scientific, respectively). sgRNA was transcribed using the MEGAscript T7 Kit (Ambion) and extracted with RNeasy Mini Kit (Qiagen) according to the manufacturer's protocol. sgRNA was diluted to a final concentration of 159.6 ng/µl or 250 ng/µl, respectively using water and 1 M KCl (final concentration: 300 mM) and co-injected with Cas9-Protein of 600 ng/µl final concentration as described (*Burger et al., 2016*; *Gagnon et al., 2014*). To determine the efficiency of sgRNA-mediated mutagenesis crispants alleles were analyzed as described (*Figure 7—figure supplement 1D*) (*Burger et al., 2016*). The following primers (BioTez Berlin-Buch GmbH) were used to amplify the genomic region flanking the CRISPR target site; forward: CAAATGGTGAACACTTGTGAATC, reverse: CTGCGGTTTACTGCGGAGATG. Computational sequence analysis was performed using CrispR Variants (*Lindsay et al., 2016*). For an injection control Cas9 was diluted to a final concentration of 600 ng/µl using water and 1 M KCl (final concentration: 300 mM; Cas9-Control).

## Reverse transcriptase (RT)-PCR in zebrafish embryos

For efficiency analysis of the *tmem63c* ex2-sdMO a reverse transcriptase (RT)-PCR was carried out (*Figure 7—figure supplement 1F*). At 24 hpf, RNA from 50 pooled embryos was isolated using Trizol Reagent (Invitrogen); DNase I digestion was performed using the RNAse-free DNase set (Qiagen) and samples were purified using the RNeasy Mini Kit (Qiagen) according to the manufacturer's protocol. After determination of RNA quality and quantity, equal amounts of mRNA for each group analyzed were transcribed to cDNA using First strand cDNA synthesis kit (Thermo Fisher) according to the manufacturer's protocol.

We amplified *tmem63c* from cDNA using DreamTaq DNA Polymerase (Thermo Fisher) with the following primers: forward: CTGATGGAGGAGAACAGCACGG, reverse: ATACAGCAGAGCGAAGATACTGTG. Eucaryotic elongation factor 1 alpha 1, like 1 (*eef1a1l1*) was used as a loading control and amplified using the following primers: forward: TGGAGACAGCAAGAACGACC, reverse: GAGGTTGGGAAGAACACGCC.

## Cloning of *tmem63c* cDNA (*Danio rerio*)

Primers (BioTez Berlin-Buch GmbH) for the In-Fusion HD Cloning Kit (Takara) were designed using the web tool provided by TaKaRa (TaKaRa) (*TaKaRa, 2018*); forward: GCTTGATATCGAATTCA TGGCGTTTGAGTCCTGGCCTGC, reverse: CGGGCTGCAGGAATTCTCACTGAAAAGCCACCGGAC TG. *tmem63c* cDNA was amplified using Phusion High-Fidelity DNA polymerase (Thermo Fisher Scientific). The pBluescript II SK(+) vector was linearized using EcoRI FD enzyme. The *tmem63c* ORF was cloned into the pBluescript II SK (+) vector using the In-Fusion HD Cloning Kit (Takara). The *tmem63c* cDNA was sequence-verified using the common T7 forward and M13 reverse primers. For sequencing of the whole ORF, an additional primer was used, GTGCAGAAACTAATGAAGCTGG, located at 822–844 bp starting from the beginning of the ORF.

## Rescue of CRISPR/Cas9-mediated *tmem63c* somatic mutants and *tmem63c* ex2-sdMO-mediated gene knockdown

For in vivo rescue experiments *tmem63c* cDNA (*Danio rerio*) was linearized using ApaI FD (Thermo Fisher Scientific) and purified using GeneJET Gel Extraction Kit (Thermo Fisher Scientific). In vitro transcription of capped RNA and following TurboDNase treatment were performed using mMessage mMachine T7 Kit (Ambion). For poly-A-tailing, the Poly(A)-tailing Kit (Ambion) was used, followed by RNA extraction using RNeasy Mini Kit (Qiagen). The mRNA was diluted to a concentration of 100 ng/µl and injected into one-cell stage zygotes. For in vivo rescue experiments, mRNA of a concentration of 100 ng/µl and ex2-sgRNA of a concentration of 159.6 ng/µl were subsequently injected into the same one-cell stage zygotes. For in vivo rescue of the *tmem63c* ex2-sdMO-mediated knockdown, mRNA with a concentration of 100 ng/µl and ex2-sdMO with a concentration of 0.5 mM were subsequently injected into the same zygote at one- or one-to-four cell stage, respectively.

## Rescue of *tmem63c* ATG-MO-mediated gene knockdown

*Tmem63c* cDNA (Rattus norvegicus) was synthesized by Thermo Fisher Scientific using their GeneArt Gene synthesis service. For in vivo rescue experiments, *Tmem63c* cDNA was linearized using XbaI FD (Thermo Fisher Scientific) and purified using GeneJET Gel Extraction Kit (Thermo Fisher Scientific). In vitro transcription of capped RNA followed by TurboDNase treatment were performed using mMessage mMachine T7 Kit (Ambion). For poly-A-tailing, the Poly(A)-tailing Kit (Ambion) was used, followed by RNA extraction by RNeasy Mini Kit (Qiagen). The mRNA was diluted to a concentration of 100 ng/µl and injected into one-cell stage zygotes. For in vivo rescue of the *tmem63c* ATG-MO-mediated knockdown, mRNA with a concentration of 100 ng/µl and ATG-MO with a concentration of 0.3 mM were subsequently injected into the same zygote at one- or one to four cell-stage, respectively.

## Functional assessment of the GFB

To assess the functionality of the GFB, the gc-EGFP fluorescence in the trunk vasculature *of Tg (fabp10a:gc-EGFP)* embryos were evaluated at 120 hpf by epifluorescence microscopy. For CRISPR-Cas9-mediated somatogenesis of *tmem63c* the above described sgRNA was used in a concentration of 159.6 ng/µl. Each embryo was visually assigned to the 'fluorescent group', 'deficient-fluorescent group', or 'crippled/dead' and their number quantified. Due to the heterogeneous genotype of the used transgenic *Tg(fabp10a:gc-EGFP)* zebrafish families, the 'deficient-fluorescent group' included the embryos with reduced fluorescence in the trunk as well as embryos that did not carry the transgene (*Figure 7—figure supplement 1A–C*). The percentage of injected embryos was normalized to the percentage of the control group for each category. Quantifications were performed for at least three individual injections.

## Electron microscopy in zebrafish embryos

*Tg(fabp10a:gc-EGFP)* embryos at 120 hpf were fixed in 4% formaldehyde/0.5% glutaraldehyde (EM-grade) in 0.1 M phosphate buffer for 2 hr at RT. For knockdown analysis embryos were injected with *tmem63c* ex2-sgRNA of a concentration of 250 ng/µl to enhance the observed phenotype (*Figure 7E,F*). Prior to analysis embryos were sorted for a clear knockdown phenotype. Samples were stained with 1% $OsO_4$ for 2 hr, dehydrated in a graded ethanol series and propylene oxide and embedded in Poly/Bed[R] 812 (Polysciences, Eppelheim, Germany). Ultrathin sections were contrasted

with uranyl acetate and lead citrate. Sections were examined with a FEI Morgagni electron microscope and a Morada CCD camera (EMSIS GmbH, Münster, Germany). Image acquisition and quantification of podocyte foot process width and number of slit diaphragms per µm GBM was performed with the iTEM software (EMSIS GmbH, Münster, Germany).

## Confocal microscopy of zebrafish embryos and quantification of podocyte cell number and glomerular volume

*Tg(wt1b:EGFP)* embryos at 96 hpf were fixed in PEM buffer containing 4% formaldehyde and 0.1% Triton-X 100 for 2 hr at RT or overnight at 4°C. For knockdown analysis embryos were injected with *tmem63c* ex2-sgRNA with a concentration of 250 ng/µl to enhance the observed phenotype. Nuclei were stained using 4′,6-Diamidin-2-phenylindol (DAPI, Sigma Aldrich, stock solution 1 mg/ml diluted 1:2000 in PBS) overnight at 4°C. After removal of the yolk and mounting in 0,7% low-melting agarose, the kidneys of whole-mount fixed embryos were imaged using a Zeiss LSM 710 or LSM 700 microscope with a LD C-Apochromat 40 x NA1.1 water objective and ZEN 2.1 software by sequentially acquiring confocal z-stacks of the GFP (488 nm laser, emission 495–550 nm) and the DAPI signal (405 nm laser, emission 420–480 nm) with a pixel size of 102.4 nm. Care was taken to apply identical settings to all samples and not to oversaturate pixels.

Quantification of podocyte cell number and glomerular volume was done using Imaris version 9.21 software (RRID:SCR_007370, Bitplane AG, Zurich, Switzerland). A 3D surface covering the total glomerular volume was manually edited by tracing the outlines of EGFP-positive cells for every second section of the z-stack. EGFP-positive cells of the glomerulus were included, while cells of the pronephric ducts were excluded. For quantification of podocyte cell number, the DAPI channel was masked with the EGFP channel using Fiji software (RRID:SCR_002285) (*Schindelin et al., 2012*) to include DAPI$^+$/EGFP$^+$ cells only, thus representing nuclei of podocyte cells. Subsequently, a spot segmentation of the DAPI channel was performed. Estimated spot diameter was 4 µm. Spots were filtered for a minimum intensity of the EGFP channel and by using the Imaris quality filter for the occurrence of unspecific spots not matching the EGFP signal. Spots located outside the glomerular surface were manually deleted.

### Statistics

Data are presented as mean ±SD for normally distributed data or median (25% percentile – 75% percentile, that is interquartile range [IQR]) for non-normally distributed data with the indicated number of experiments. Normal distribution was determined using the Shapiro-Wilk test. For identification of outliers, Grubbs' outliers test ($\alpha$ = 0.05) was performed. Where appropriate, sample size calculations were performed by the power analyses program G*Power according to Cohen (*Cohen, 1988*; *Faul et al., 2009*). Differences between experimental rat and zebrafish groups were analyzed using One-way ANOVA with post-hoc Bonferroni's multiple comparisons test and non-parametric Mann-Whitney-U or Kruskall-Wallis test with Dunn's multiple comparisons post-hoc test, when appropriate. For the analysis shown in *Figure 7L* and *Figure 7—figure supplement 1I* Gaussian distribution was assumed due to the number of embryos categorized. Differences between FSGS patients and controls were analyzed using the Linear-by-Linear association test and the Mann-Whitney U test. Differences in human cultured podocytes were analyzed using two-tailed Student's t-test. Statistical analysis was performed using SPSS and GraphPad Prism 6 software 6.00 (RRID:SCR_002798, GraphPad Software, La Jolla, CA). p values < 0.05 were considered as statistically significant.

### Data availability

The genomic and transcriptomic data from this publication have been deposited to the NCBI (https://www.ncbi.nlm.nih.gov/) curated repositories, GEO, and SRA, and assigned the identifier SubmissionID: SUB2950675 and BioProject ID: PRJNA398197 (DNA-Seq) and accession GSE102546 (RNA-Seq).

## Acknowledgements

We acknowledge the contributions of Karen Böhme, Bettina Bublath, Marianne Jansen-Rust, Claudia Plum, Christiane Priebsch, Christina Schiel, Sabine Wunderlich[†], and Malu Zandbergen for excellent laboratory or animal assistance. *Tg(wt1b:GFP)* line was kindly provided by Christoph Englert. This

study was supported by the Deutsche Hochdruckliga (DHL) Hypertensiologie Professur to RK, by the grants from the Deutsche Forschungsgemeinschaft (DFG) KR 1152-3-1 (RK) and SCHU 2604/1–1 (ASch), by the DFG (German Research Foundation) – Project number 394046635 – SFB 1365 (RK, DP), and by the grants from the Helmholtz Young Investigator Program VH-NG-736, and Marie Curie CIG from the European Commission (WNT/CALCIUM IN HEART-322189) (DP).

## Additional information

### Funding

| Funder | Grant reference number | Author |
|---|---|---|
| Deutsche Forschungsge-meinschaft | DFG KR 1152-3-1 | Reinhold Kreutz |
| Helmholtz-Gemeinschaft | VH-NG-736 | Daniela Panáková |
| European Commission | WNT/CALCIUM IN HEART-322189 | Daniela Panáková |
| Deutsche Forschungsge-meinschaft | SCHU 2604/1-1 | Angela Schulz |
| Deutsche Forschungsge-meinschaft | Project number 394046635 - SFB 1365 | Reinhold Kreutz |
| Deutsche Hochdrckliga (DHL) | Stiftungsprofessur Hypertensiologie | Reinhold Kreutz |

The funders had no role in study design, data collection and interpretation, or the decision to submit the work for publication.

### Author contributions

Angela Schulz, Conceptualization, Resources, Data curation, Formal analysis, Supervision, Funding acquisition, Validation, Investigation, Visualization, Methodology, Writing—original draft, Project administration, Writing—review and editing; Nicola Victoria Müller, Nina Anne van de Lest, Conceptualization, Data curation, Formal analysis, Supervision, Validation, Investigation, Visualization, Methodology, Writing—original draft, Writing—review and editing; Andreas Eisenreich, Conceptualization, Data curation, Formal analysis, Supervision, Validation, Investigation, Visualization, Methodology, Writing—original draft, Project administration, Writing—review and editing; Martina Schmidbauer, Conceptualization, Data curation, Formal analysis, Validation, Investigation, Visualization, Methodology, Writing—original draft, Writing—review and editing; Andrei Barysenka, Resources, Data curation, Formal analysis, Validation, Investigation, Visualization, Writing—review and editing; Bettina Purfürst, Resources, Data curation, Formal analysis, Validation, Methodology, Writing—original draft, Writing—review and editing; Anje Sporbert, Resources, Methodology, Writing—review and editing; Theodor Lorenzen, Investigation, Writing—review and editing; Alexander M Meyer, Laura Herlan, Investigation, Methodology, Writing—review and editing; Anika Witten, Formal analysis, Validation, Investigation, Methodology, Writing—review and editing; Frank Rühle, Resources, Formal analysis, Validation, Investigation, Visualization, Writing—original draft, Writing—review and editing; Weibin Zhou, Resources, Writing—review and editing; Emile de Heer, Conceptualization, Resources, Validation, Writing—review and editing; Marion Scharpfenecker, Conceptualization, Supervision, Validation, Methodology, Writing—original draft, Project administration, Writing—review and editing; Daniela Panáková, Reinhold Kreutz, Conceptualization, Resources, Data curation, Supervision, Funding acquisition, Validation, Methodology, Writing—original draft, Project administration, Writing—review and editing; Monika Stoll, Conceptualization, Data curation, Software, Supervision, Validation, Methodology, Writing—original draft, Project administration, Writing—review and editing

### Author ORCIDs

Angela Schulz http://orcid.org/0000-0002-4576-8035
Nicola Victoria Müller http://orcid.org/0000-0001-7261-830X

Daniela Panáková (iD) https://orcid.org/0000-0002-8739-6225
Reinhold Kreutz (iD) http://orcid.org/0000-0002-4818-211X

## Ethics

Human subjects: All biopsy samples were handled and analyzed anonymously in accordance with the Dutch National Ethics Guidelines (Code for Proper Secondary Use of Human Tissue, Dutch Federation of Medical Scientific Societies). Because this study concerned retrospectively collected anonymized material, no informed consent was necessary following the Dutch National Ethics Guidelines. This study is in agreement with the Declaration of Helsinki and the Department of Health and Human Services Belmont Report and the use of the patient biopsies was approved by the medical ethical committee of the LUMC (registration number G16.110).

Animal experimentation: All experimental work in rat models was performed in accordance with the guidelines of the Charité-Universitätsmedizin Berlin and the local authority for animal protection (Landesamt für Gesundheit und Soziales, Berlin, Germany) for the use of laboratory animals. The registration numbers for the rat experiments are G 0255/09 and T 0189/02. Zebrafish were bred, raised and maintained in accordance with the guidelines of the Max Delbrück Center for Molecular Medicine and the local authority for animal protection (Landesamt für Gesundheit und Soziales, Berlin, Germany) for the use of laboratory animals, and followed the 'Principles of Laboratory Animal Care' (NIH publication no. 86-23, revised 1985) as well as the current version of German Law on the Protection of Animals.

## Decision letter and Author response

Decision letter https://doi.org/10.7554/eLife.42068.039
Author response https://doi.org/10.7554/eLife.42068.040

# Additional files

## Supplementary files

• Transparent reporting form
DOI: https://doi.org/10.7554/eLife.42068.033

## Data availability

The genomic and transcriptomic data from this publication have been deposited to the NCBI curated repositories, GEO, and SRA, and assigned the identifier SubmissionID: SUB2950675 and BioProject ID: PRJNA398197 (DNA-Seq) and accession GSE102546 (RNA-Seq).

The following datasets were generated:

| Author(s) | Year | Dataset title | Dataset URL | Database and Identifier |
|---|---|---|---|---|
| Barysenka A, Schulz A, van de Lest NA, Eisenreich A, Schmidbauer M, Müller N, Lorenzen T, Meyer A, Herlan L, Witten A, Rühle F, Zhou W, de Heer E, Scharpfenecker M, Panakova D, Stoll M, Kreutz R | 2017 | Integrative analysis of the genomic architecture of a complex kidney damage QTL in inbred hypertensive rats implies Tmem63c for translational research | https://www.ncbi.nlm.nih.gov/bioproject/?term=PRJNA398197 | NCBI BioProject, PRJNA398197 |
| Barysenka A, Schulz A, van de Lest NA, Eisenreich A, Schmidbauer M, Müller N, Lorenzen T, Meyer A, Herlan L, Witten A, Rühle F, Zhou W, de Heer E, Scharpfenecker M, Panakova D, | 2017 | Integrative analysis of the genomic architecture of a complex kidney damage QTL in inbred hypertensive rats implies Tmem63c for translational research | https://www.ncbi.nlm.nih.gov/geo/query/acc.cgi?acc=GSE102546 | NCBI Gene Expression Omnibus, GSE102546 |

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
