## [Decision Letter]

Thank you for submitting your article "Analysis of the genomic architecture of a complex trait locus in hypertensive rat models links *Tmem63c* to kidney damage" for consideration by *eLife*. Your article has been reviewed by two peer reviewers, and the evaluation has been overseen by a Reviewing Editor and Mark McCarthy as the Senior Editor. The reviewers have opted to remain anonymous.

The reviewers have discussed the reviews with one another and the Reviewing Editor has drafted this decision to help you prepare a revised submission.

The manuscript describes mapping and putative identification of a QTL gene for rat albuminuria with subsequent experiments in zebrafish and human glomerulosclerosis indicating a role in mediating glomerular filtration and podocyte function. Whilst the work has been competently carried out and the manuscript is well written, reviewers have raised concerns that need to be addressed before considering publication in the journal. In particular, to demonstrate specificity, rescue of the morpholino injected zebrafish with *tmem63c* cRNA should be carried out. We are also keen to see results from the IF experiments listed in the reviewer responses below (though we do not see this as essential as the morpholino rescue). Other comments of the reviewers should be addressed in point-by-point responses.

Reviewer #1:

Dr. Schulz et al. conducted the comprehensive analysis of a major albuminuria susceptibility locus detected in inbred rat models of kidney damage associated with elevated blood pressure. They found the TMEM63C loss in podocytes of FSGS patients, as well as possible role of *tmem63c* in mediating the glomerular filtration barrier function in zebrafish model. However, the following concerns should be raised.

1) TMEM63C expression in human kidney: Are there any previous findings to demonstrate that TMEM63C is expressed in human kidney?

2) Other genes expression of causative genes in FSGS patients: How is the expression of other causative genes for FSGS such as ACTN4, TRCP6, PAX2 etc. in FSGS patients? Also could you provide the potential interaction between TMEM63C. Please provide the explanation for the possible interaction between TMEM63C and the known causative genes for FSGS.

*Reviewer #2:*

The authors convincingly show that transmembrane protein *Tmem63c* is a genetic component that can underlie albuminuria in rats by comparing the MWF rat and SHR rat models in a QTL trait mapping and target validation strategy. Following QTL, in the region of interest, the authors performed next generation sequencing analysis and assembled a list of gene candidates. To further identify relevant candidates as relates to kidney damage, they performed RNA-seq on glomeruli isolated from the rat models and compared transcriptional profiles between these models. They ultimately narrowed the gene list to 10 differentially expressed genes, and from this selected *Tmem63c*. They also found that patients with focal glomerulosclerosis have TMEM63C loss in podocytes and nicely used siRNA to determine the effect of reduced TMEM63C in human podocytes in culture. Finally, they performed loss of function studies in the zebrafish and present the findings that *tmem63c* is necessary for an intact glomerular filtration barrier. Overall, this is a very well written and presented study.

1) The authors show nicely that they can rescue the zebrafish *tmem63c* crispants with *tmem63c* cRNA. The ATG morpholino injected animals need to be assessed similarly for rescue, to demonstrate specificity of the morpholino reagent.

2) The authors should assess what happens to podocytes in the *tmem63c* morphants or crispants by whole mount in situ hybridization or immunofluorescence studies at stages after edema has developed, to determine whether proteinuria is associated with podocyte loss. The TEM images currently suggest GBM disruption, but quantification of podocyte retention on the GBM would improve the manuscript in this aspect.

---

## [Author Response]

Reviewer #1:1) TMEM63C expression in human kidney: Are there any previous findings to demonstrate that TMEM63C is expressed in human kidney?

To the best of our knowledge, there are no previous reports in the literature that demonstrate TMEM63C expression in the human kidney.

Assessment of the human protein atlas database (https://www.proteinatlas.org/) indicates TMEM63C expression in the human kidney both in the glomerular and tubular compartment (https://www.proteinatlas.org/ENSG00000165548-TMEM63C/tissue/kidney#img).

2) Other genes expression of causative genes in FSGS patients: How is the expression of other causative genes for FSGS such as ACTN4, TRCP6, PAX2 etc. in FSGS patients? Also could you provide the potential interaction between TMEM63C.

Analysis of other causative genes in FSGS patients:

We realize that the exploration of other causative genes in FSGS patients appears intriguing. However, such analysis is challenging for several reasons: i) the increasing number of genes (>50) that are currently implicated in FSGS (De Vriese, Sethi, Nath, Glassock, and Fervenza, 2018; Lepori, Zand, Sethi, Fernandez-Juarez, and Fervenza, 2018); ii) the limited material of renal biopsy samples available; iii) and the limited number of antibodies, e.g. for PAX2, available for reliable immunohistochemistry analysis in human biopsy tissues. Furthermore, changes in expression may differ depending on the selected genes. For example, TRPC6 protein has been shown to be markedly upregulated in carriers of TRPC6 mutations but can be unchanged in patients with FSGS and proteinuria without TRPC6 mutations (Gigante et al., 2011). Thus, a conclusive analysis would require a comprehensive genotype-phenotype analysis (De Vriese et al., 2018; Lepori et al., 2018).

Nevertheless, we agree that determining the expression of selected podocyte related proteins would be insightful and in response to the reviewer´s comment, we set out to analyze expression of the slit pore protein nephrin (NPHS1) and the cytoskeleton protein ACTN4 in our set of FSGS patients. Mutations in both of these genes are known to cause FSGS (Kaplan et al., 2000; Kestila et al., 1998). Furthermore, reduced protein expression of both nephrin and ACTN4 has been shown in acquired forms of FSGS as well (Kim, Hong, Kim, and Lee, 2002; Testagrossa, Azevedo Neto, Resende, Woronik, and Malheiros, 2013). First, we investigated the expression of nephrin in our cohort of patients with FSGS. We found that nephrin expression was significantly reduced in parallel to TMEM63C in our set of patients with FSGS. This change of nephrin expression is in accordance with previously published results (Kim et al., 2002). Moreover, we observed a shift from the normal linear staining pattern to a granular staining pattern also in agreement with previous reports (Doublier et al., 2001; Wernerson et al., 2003). Although there is little known about the interaction between TMEM63C and nephrin, we can still speculate about a link between these two proteins: nephrin is involved in mechanosensing and integration of the slit pore with the mechanotransduction machinery of the podocyte and TMEM63C has originally been suggested as a mechanosensitive ion channel (Hou et al., 2014; Zhao, Yan, Liu, Zhang, and Ni, 2016). Therefore, both molecules could be involved in mechanosensing of the podocyte.

Next, we set out to investigate the expression of the cytoskeleton protein ACTN4. ACTN4 is involved in mechanotransduction as well (Feng et al., 2018) and could thus be functionally linked to TMEM63C. However, when we stained a cohort of healthy control subjects for ACTN4 (Santa Cruz Biotechnology), we observed a heterogeneous staining pattern and some cases did not show any staining. Staining of another cohort of healthy controls resulted in the same heterogeneous staining pattern. This is not in accordance with previously published results on ACTN4 expression in the normal glomerulus (Liu et al., 2011; Testagrossa et al., 2013). We therefore omitted the immunostaining for ACTN4 in patients with FSGS.

The new data on nephrin staining in FSGS patients are presented in revised Figure 5 together with the TMEM63c analysis.

The data obtained in podocytes in cell culture in Figure 5G-J of the original manuscript are presented in the new Figure 6 of the revised manuscript (because otherwise the new Figure 5 would be overloaded).

The text of the revised manuscript has been adapted with the corresponding information in the Results section:

“In addition to TMEM63C expression, we analyzed the expression of nephrin protein as a pivotal component of the slit diaphragm of the GFB (Figure 5G-J) (Kestila et al., 1998). […] Moreover, we observed a shift from the normal linear staining pattern to a granular staining pattern as reported (Figure 5H, J) (Doublier et al., 2001; Wernerson et al., 2003).”

In the Discussion section:

“Furthermore, the loss of TMEM63C was associated with a decrease and altered granular staining pattern of nephrin protein in glomeruli of FSGS patients as previously reported (Doublier et al., 2001; Wernerson et al., 2003). […] While our data cannot establish a functional link between changes in nephrin and TMEM63c in FSGS, they nonetheless demonstrate a concomitant deficiency of both proteins in this setting.”

In the Materials and methods section:

“For nephrin protein expression analysis we used a rabbit anti-nephrin antibody (Abcam). Paraffin embedded rat kidney sections and human biopsy samples were cut at 4 µm and incubated with the anti-TMEM63C antibody (1:1600 for rat tissue, 1:800 for human biopsies), the anti-WT1 antibody (1:500) or the anti-nephrin antibody (1:750).”

“For nephrin staining analysis, each glomerulus was scored based on staining pattern (linear or granular) and loss of staining (no loss, segmental loss or global loss).”

Please provide the explanation for the possible interaction between TMEM63C and the known causative genes for FSGS.

As indicated above multiple genes have been implicated to play a causative role or at least modify the susceptibility for the development of FSGS (De Vriese et al., 2018; Lepori et al., 2018; Yu et al., 2016). As indicated in the original manuscript, the function of the majority of the more than 300 TMEM proteins is currently unclear (Wrzesinski et al., 2015). The mechanism by which changes in *Tmem63c* expression cause podocyte damage and thus contribute to the final common pathway of injury in FSGS remains also unclear and cannot be elucidated by our current analysis. Accordingly, our data do not describe possible interactions between TMEM63c and the heterogeneous group of known causative FSGS genes. Nevertheless, previous reports suggested that T*mem63c* and two other genes of the *tmem63*-family, e.g. *Tmem63a* and *Tmem63b,* are mammalian homologues of the hyperosmolarity-activated cation channel proteins AtCSC1 and its paralogue OSCA1 in plants (Hou et al., 2014; Zhao et al., 2016). Based on these reports it appeared tempting to speculate, that *Tmem63c* may be involved in mechanosensing in podocytes providing thus a functional basis for interactions with cytoskeleton genes, e.g. ACTN4 or slit pore proteins, e.g. nephrin, in podocytes. However the potential role of *Tmem63c* as a hyperosmolarityactivated cation channel remains controversial, since the activation by hyperosmolarity was not confirmed in a more recent study (Murthy et al., 2018). Moreover, the latter report indicated that *Tmem63c* is phylogenetically divergent from *Tmem63a* and *Tmem63b* and lacks their function to induce stretch-activated ion currents. Consequently, as indicated in the original manuscript further studies are needed to characterize the functional role of *Tmem63c* and to explore its potential to influence podocyte function as a novel target for therapeutic intervention (Endlich, Kliewe, and Endlich, 2017; Forst et al., 2016; Wieder and Greka, 2016).

In response to the reviewer´s comment revised the text in the Discussion as follows:

“The mechanism by which changes in *Tmem63c* expression cause podocyte damage and thus contribute to the final common pathway of injury in FSGS (De Vriese et al., 2018) remains however unclear and cannot be elucidated by our current analysis. […] Moreover, the latter report indicated that *Tmem63c* is phylogenetically divergent from *Tmem63a* and *Tmem63b* and lacks their function to induce stretch-activated ion currents.”

Reviewer #2:1) The authors show nicely that they can rescue the zebrafish tmem63c crispants with tmem63c cRNA. The ATG morpholino injected animals need to be assessed similarly for rescue, to demonstrate specificity of the morpholino reagent.

We addressed this important issue as requested and performed rescue experiments using rat *Tmem63c* mRNA for the ATG morpholino injected zebrafish embryos. These experiments were carried out by co-injection of rat *Tmem63c* mRNA with *tmem63c* ATG-MO. In summary, we not only show the specificity of the observed knockdown phenotype in both crispants and morphants, but also conservation of the gene’s function across species. The new data on rescue experiments are presented in revised Figure 7I and L. For better visual inspection, the presentation in panel L was also adapted by showing in the revision the fluorescence deficiency rather than the presence of fluorescence as in the original submission.

The text of the revised manuscript has been adapted with the corresponding information in the Introduction:

“Loss-of-function studies in zebrafish induced a glomerular filtration barrier (GFB) defect compatible with the albuminuria phenotype, which was rescued upon co-injection of zebrafish *tmem63c* mRNA or rat *Tmem63c* mRNA, showing not only the specificity of the observed knockdown phenotype, but also conservation of the gene’s function across species.”

In the Results section:

“To verify the specificity of the observed phenotype rescue experiments were carried out by co-injection of zebrafish *tmem63c* mRNA with *tmem63c* sgRNA/Cas9 complexes and rat *Tmem63c* mRNA (mRNA sequence identity vs. zebrafish = 65.82% (Clustal 2.1), Figure 7—figure supplement 2) with *tmem63c* ATG-MO, respectively. For both cases similarly, the albuminuria-like phenotype could be specifically rescued proving knockdown specificity on the one hand and functional conservation of *tmem63c* across species on the other hand (Figure 7I, K and L). “

In the Discussion section:

“Importantly, rescue of the observed phenotypes in zebrafish with rat *Tmem63c* mRNA pointed to the functional conservation across species.”

In the Materials and methods section:

“Rescue of *tmem63c* ATG-MO-mediated gene knockdown.

*Tmem63c* cDNA (Rattus norvegicus) was synthesized by Thermo Fisher Scientific using their GeneArt Gene synthesis service. […] The mRNA was diluted to a concentration of 100 ng/μl and injected into one-cell stage zygotes. For in vivo rescue of the *tmem63c* ATG-MO-mediated knockdown, mRNA with a concentration of 100 ng/µl and ATG-MO with a concentration of 0.3 mM were subsequently injected into the same zygote at one- or one to four cell-stage, respectively.”

In addition we performed also similar rescue experiments in the ex2-sdMO, i.e. the splicing site morpholino, gaining a similar knockdown phenotype and positive rescue results as shown in Figure 7—figure supplement 1. To prove the efficiency of the *tmem63c* ex2-sdMO an additional reverse transcriptase (RT) – PCR was carried out as shown in Figure 7—figure supplement 1F.

The text of the revised manuscript has been adapted with the corresponding information in the Results section:

“We corroborated this finding in *tmem63c* crispants (Figure 7J, Figure 7—figure supplement 1D) as well as by using another splice-blocking morpholino (Figure 7—figure supplement 1E, I); both experiments showed a similar albuminuria-like phenotype.”

In the Materials and methods section:

“Morpholinos (MO) of the following sequences were synthesized by Gene Tools LLC Philomath: *tmem63c* ATG-MO 5’-CAGGCCAGGACTCAAACGCCATTGC-3’, *tmem63c* ex2sdMO 5'-TGTTATCATAGATGATGTACCAGCC-3', and standard control oligo (Control-MO) 5’-CCTCTTACCTCAGTTACAATTTATA-3’. *Tmem63c* ATG-MO was used at a final concentration of 0.3 mM (Figure 7B), *tmem63c* ex2-sdMO was used at a final concentration of 0.5 mM (Figure 7—figure supplement 1).”

**“**Reverse transcriptase (RT)-PCR in zebrafish embryos

For efficiency analysis of the *tmem63c* ex2-sdMO a reverse transcriptase (RT)-PCR was carried out (Figure 7—figure supplement 1F). […] Eucaryotic elongation factor 1 α 1, like 1 *(eef1a1l1)* was used as a loading control and amplified using the following primers: forward: TGGAGACAGCAAGAACGACC, reverse: GAGGTTGGGAAGAACACGCC.”

“For in vivo rescue of the *tmem63c* ex2-sdMO-mediated knockdown, mRNA with a concentration of 100 ng/µl and ex2-sdMO with a concentration of 0.5 mM were subsequently injected into the same zygote at one- or one to four cell-stage, respectively.”

2) The authors should assess what happens to podocytes in the tmem63c morphants or crispants by whole mount in situ hybridization or immunofluorescence studies at stages after edema has developed, to determine whether proteinuria is associated with podocyte loss. The TEM images currently suggest GBM disruption, but quantification of podocyte retention on the GBM would improve the manuscript in this aspect.

As suggested, we performed immunofluorescence studies at 96 hours post fertilization (hpf), i.e. the time after development of edema, in crispants. To analyze, whether the observed albuminuria-like phenotype upon *tmem63c*-deficiency is associated with the loss of podocytes, we used confocal microscopy to image glomeruli of Tg(wt1b:EGFP) embryos. We employed the transgenic *Tg(wt1b:EGFP*) zebrafish line, because it allows specific visualization of the pronephros and is thus suitable for this analysis.(Perner, Englert, and Bollig, 2007).

In this analysis, we first observed that *tmem63c* crispants (*tmem63c* ex2-sgRNA) exhibit a widened Bowman´s space and increased glomerular volumes compared to uninjected controls (133457 ± 59547 µm^3^ vs 67067 ± 21933 µm^3^, P = 0.04) as quantified by 3D surface reconstruction. In addition, we observed dilated capillary loops in the crispants. The subsequent quantification of podocyte cell number revealed no changes in absolute cell number, while relative podocyte cell number normalized to the total glomerular volume was significantly decreased compared to uninjected controls.

The new data are presented in revised Figure 8 of the revised manuscript. In this regard, we also feel that it would be more appropriate to show the structural analysis of glomeruli together in one figure. Consequently, we added the TEM results, which we showed in Figure 6 of the original manuscript to the revised Figure 8. In addition, Videos 1-3 are supplied together with the revised manuscript to further illustrate the software-based 3D analysis done in confocal z-stacks.

The text of the revised manuscript has been adapted with the corresponding information in the Results section:

“To analyze, whether the observed albuminuria-like phenotype upon *tmem63c*-deficiency is associated with the loss of podocytes, we utilized confocal microscopy to image glomeruli of *Tg(wt1b:EGFP)* embryos. […] Collectively, our data indicate the conserved role for *tmem63c* in GFB function between fish, rodents, and humans*.”*

In the Discussion section:

“Further structural analysis by confocal microscopy revealed dilated capillary loops and enlarged glomerular volumes with a reduction of relative podocyte number in relation to glomerular volume (podocyte density) in *tmem63c*-deficient embryos. Thus, our data reveal an important role of *tmem63c* for normal development of capillary structure and podocyte function of the glomerulus in zebrafish.”

In the Materials and methods section:

**“**Confocal microscopy of zebrafish embryos and quantification of podocyte cell number and glomerular volume.

*Tg(wt1b:EGFP)* embryos at 96 hpf were fixed in PEM buffer containing 4% formaldehyde and 0.1% Triton-X 100 for 2 hours at RT or overnight at 4°C. […] Spots located outside the glomerular surface were manually deleted.”

Video legend:

**“**Video 1-3. Representative spot segmentation of podocyte nuclei and 3D surface reconstruction of glomeruli in *Tg(wt1b:EGFP)* zebrafish embryos.[…] Shown here are representative analysis results of uninjected controls (Video 1), Cas9-Controls (Video 2) and *tmem63c* ex2-sgRNA-injected embryos (Video 3).”